# The interactome of the copper transporter ATP7A belongs to a network of neurodevelopmental and neurodegeneration factors

Heather S Comstra[1†], Jacob McArthy[2†], Samantha Rudin-Rush[3†], Cortnie Hartwig[3†], Avanti Gokhale[1†], Stephanie A Zlatic[1†], Jessica B Blackburn[4], Erica Werner[5], Michael Petris[6], Priya D'Souza[7], Parinya Panuwet[7], Dana Boyd Barr[7], Vladimir Lupashin[4], Alysia Vrailas-Mortimer[2*], Victor Faundez[1*]

[1]Departments of Cell Biology, Emory University, Atlanta, United States; [2]School of Biological Sciences, Illinois State University, Normal, United States; [3]Department of Chemistry, Agnes Scott College, Decatur, Georgia; [4]Department of Physiology and Biophysics, University of Arkansas for Medical Sciences, Little Rock, United States; [5]Department of Biochemistry, Emory University, Atlanta, United States; [6]Department of Biochemistry, University of Missouri, Columbia, United States; [7]Rollins School of Public Health, Emory University, Atlanta, United States

*For correspondence: avraila@ gmail.com (AV-M); vfaunde@ emory.edu (VF)

†These authors contributed equally to this work

Competing interests: The authors declare that no competing interests exist.

**Abstract** Genetic and environmental factors, such as metals, interact to determine neurological traits. We reasoned that interactomes of molecules handling metals in neurons should include novel metal homeostasis pathways. We focused on copper and its transporter ATP7A because ATP7A null mutations cause neurodegeneration. We performed ATP7A immunoaffinity chromatography and identified 541 proteins co-isolating with ATP7A. The ATP7A interactome concentrated gene products implicated in neurodegeneration and neurodevelopmental disorders, including subunits of the Golgi-localized conserved oligomeric Golgi (COG) complex. COG null cells possess altered content and subcellular localization of ATP7A and CTR1 (SLC31A1), the transporter required for copper uptake, as well as decreased total cellular copper, and impaired copper-dependent metabolic responses. Changes in the expression of ATP7A and COG subunits in *Drosophila* neurons altered synapse development in larvae and copper-induced mortality of adult flies. We conclude that the ATP7A interactome encompasses a novel COG-dependent mechanism to specify neuronal development and survival.

## Introduction

Copper, iron, and manganese act as micronutrients, yet in excess behave as environmental neuro-toxicants (*Bush, 2013*; *Perl and Olanow, 2007*; *Wright and Metals, 2007*; *Madsen and Gitlin, 2007*). These metals are handled by diverse membrane transporters and dedicated chaperones that deliver metals to cellular compartments to act as cofactors in essential enzymatic reactions while also maintaining cellular levels of these metals within a narrow range (*Robinson and Winge, 2010*). Mutations in these metal membrane transport and chaperoning mechanisms cause neurodegenerative disorders in both children and adults (*Madsen and Gitlin, 2007*; *Kaler, 2011*). In addition, metal exposure modulates the severity of neurological disease by yet unknown metal-sensitive mechanisms

**eLife digest** People need a source of copper in their diet because this nutrient is used to produce the pigment in hair and skin, the connective tissue in tendons and ligaments, and some of the small molecules that allow brain cells to communicate. There is an ideal range of copper that allows cells to carry out these processes. Both too much and too little copper can have negative effects on health, particularly related to how the brain works.

Cells contain multiple proteins that bind to copper and transport it wherever it is needed. People with mutations that mean they lack one of these copper transporters, ATP7A, often have serious damage to their nervous system that cannot be explained by the current understanding of how this protein works.

Comstra et al. set out to establish a comprehensive list of proteins that interact with ATP7A to better understand how this transporter works and how it is regulated. The search revealed that ATP7A interacts with hundreds of proteins present in different compartments within cells, many of which had not previously been associated with balancing copper levels in cells and the body. Like ATP7A, many of these proteins (or the protein complexes that contain them) are known to affect nerves and brain activity when they are mutated.

Next, Comstra et al. engineered human cells grown in the laboratory to lack one of the protein complexes that interacts with ATP7A, the COG complex. Cells without this protein complex had 50% less ATP7A than normal human cells and very low levels of copper too. These mutant cells also had problems generating the energy that they need, because the structures in cells that provide them with energy – the mitochondria – were impaired; adding copper to the cells improved the activity of their mitochondria.

Mutations in the COG complex cause the brain to develop abnormally, and the finding that deleting the COG complex from cells causes copper deficiency now helps to explain why. Further characterization of the proteins that interact with ATP7A and the COG complex will contribute to our understanding of how cells regulate copper and how copper levels affect the brain.

(*Sparks and Schreurs, 2003*). Thus, genetic and environmental factors, such as metals, converge to impinge on neurodegeneration and neurodevelopmental phenotypes.

Cellular copper content is regulated by two chief transporters, ATP7A and CTR1 (SLC31A1) (*Madsen and Gitlin, 2007*; *Kaler, 2011*). Cellular copper homeostasis is maintained by virtue of ATP7A and CTR1 subcellular localization and metal transport topology. ATP7A resides at the Golgi apparatus where it sequesters copper topologically into the Golgi lumen and away from the cytoplasm. In contrast, CTR1 localizes to the plasma membrane where it transports copper into the cytoplasm from the extracellular milieu (*Kaler, 2011*; *Lutsenko et al., 2007*; *Polishchuk and Lutsenko, 2013*). The subcellular localization of ATP7A and CTR1 is modulated by copper-dependent vesicle traffic. After an extracellular copper challenge, ATP7A translocates from the Golgi complex to the plasma membrane where it extrudes copper out of cells while CTR1 undergoes copper-dependent endocytosis, thus down-regulating CTR1-dependent copper transport across the plasma membrane (*Kaler, 2011*; *Lutsenko et al., 2007*; *Polishchuk and Lutsenko, 2013*). Of these two transporters, ATP7A is associated with neurological phenotypes in vertebrates (*Kaler, 2011*; *Menkes, 1999*; *Menkes et al., 1962*; *Kennerson et al., 2010*; *Tümer, 2013*; *Kaler et al., 1994*), justifying it as our choice to test the hypothesis that interactomes of molecules specialized in the handling of metals encompass novel metal homeostasis mechanisms capable of modulating the expression of neurological traits.

ATP7A loss-of-function genetic defects cause three neurological diseases, including Menkes disease (OMIM 309400), occipital horn syndrome (OMIM 304150), and X-linked distal spinal muscular atrophy type 3 (OMIM 300489). Menkes causing mutations systemically deplete organisms and cells of copper due to defective gut copper uptake, which in turn reduces the activity of cuproenzymes involved in neurotransmitter, neuropeptide, melanin, mitochondrial respiration, and extracellular matrix synthesis (*Kaler, 2011*; *Lutsenko et al., 2007*; *Menkes, 1999*; *Menkes et al., 1962*; *Kennerson et al., 2010*; *Tümer, 2013*; *Kaler et al., 1994*). Menkes disease is dominated by early

childhood neurodegeneration whose mechanisms remain poorly understood (*Zlatic et al., 2015*), thus making ATP7A an ideal candidate to identify metal homeostasis mechanisms capable of modulating neuropathology phenotypes.

Here, we defined the ATP7A interactome which was enriched in products implicated in neurodegeneration and neurodevelopmental disorders, including the conserved oligomeric Golgi (COG) complex. COG is a Golgi vesicle tether necessary for intra Golgi vesicular traffic and the retention of Golgi-localized enzymes at the Golgi complex, such as glycosyltransferases (*Climer et al., 2015*; *Willett et al., 2013*; *Miller and Ungar, 2012*; *Ungar et al., 2002*; *Kranz et al., 2007*). COG genetic defects cause disruption of the Golgi complex function characterized by the loss of Golgi-resident enzymes and alterations in the glycosylation of membrane proteins traversing the exocytic route. These COG mutations cause a systemic disorder that includes cerebral atrophy, developmental delay, hypotonia, ataxia and epilepsy (*Climer et al., 2015*; *Willett et al., 2013*; *Miller and Ungar, 2012*; *Ungar et al., 2002*; *Kranz et al., 2007*). A requirement for the COG complex in cellular copper homeostasis and ATP7A-dependent metal buffering mechanisms was previously unrecognized and it has become the focus of our attention.

## Results

### The ATP7A interactome

In order to identify ATP7A-dependent and copper-sensitive mechanisms, we developed an unbiased approach to comprehensively define the ATP7A interactome in neuroblastoma cells (*Figure 1*). We used human SH-SY5Y neuroblastoma cells to perform immunoaffinity chromatography isolation of cross-linked ATP7A complexes. ATP7A was isolated with a monospecific monoclonal antibody that robustly recognizes a band in SH-SY5Y cells whose immunoreactivity is abolished in ATP7A-null cells (*Figure 1A*, lanes 1–4 and *Figure 1B*). Like other cells, SH-SY5Y ATP7A surface content increased after copper addition as determined by surface biotinylation followed by streptavidin precipitation and immunoblot against ATP7A. Transferrin receptor, a copper-insensitive membrane protein, did not increase at the surface upon copper addition (*Figure 1*, compare lanes 1' and 4'). These results validate our choice of cells and antibody to define the ATP7A interactome.

Intact cells were first cross-linked with dithiobis(succinimidyl propionate) to identify ATP7A protein interaction partners with weak or transient interactions (DSP, *Figure 1C*). DSP is a short arm 12 Å cross-linker, which is cell permeant and cleavable by reducing agents (*Alloza et al., 2004*; *Lomant and Fairbanks, 1976*; *Zlatic et al., 2010*). Cross-linking increases the coverage of interactome components co-purifying with membrane proteins of complex topology, such as ATP7A (*Gokhale et al., 2012*; *Perez-Cornejo et al., 2012*). We developed a multipronged approach to maximize ATP7A-specific interactions. First, non-selective binding to magnetic bead-ATP7A antibody complexes was determined by the addition of an excess of the antigenic ATP7A peptide recognized by the monoclonal antibody (*Figure 1C1–2* and *Figure 2B*). Second, the ATP7A antigenic peptide was used to selectively elute cross-linked ATP7A interacting proteins from magnetic beads instead of Laemmli sample buffer (*Figures 1C1–4*). Third, we mock isolated ATP7A from ATP7A-null human fibroblasts (ATP7A$^{-/-}$), along with the same cells rescued by expression of recombinant ATP7A (*Figures 1C3–4*, ATP7A$^{R/R}$) (*La Fontaine et al., 1998*). Proteins isolated from ATP7A-null cells were eliminated from all experimental datasets (*Figure 1C3*). Fourth, we performed label free quantitative mass spectrometry and thresholded positive protein identification as a ratio >2 between samples immunoisolated in the absence and presence of the ATP7A antigenic peptide (*Figures 1C* and *2A*) (*Gokhale et al., 2016*). Finally, we isolated cross-linked ATP7A complexes from SH-SY5Y neuroblastoma cells preincubated in the presence of either CuCl$_2$ or the copper chelator bathocuproine disulphonate (*Figures 1C1–2*, BCS). *Figure 1D* depicts immunoaffinity chromatography experiments with cell extracts from SH-SY5Y cells pre-treated in the absence (*Figure 1D*, BCS) or presence of copper (*Figure 1D*). Non-specific interactors bound to ATP7A antibody-decorated magnetic beads are depicted in lanes 3 and 3' where incubations were performed with an excess of the ATP7A antigenic peptide (*Figure 1D*). The presence of ATP7A and one previously described interactor, dopamine beta hydroxylase (DBH), was confirmed by immunoblot of isolated cross-linked complexes (*Figure 1D*) (*Gokhale et al., 2015a*). Protein complexes copurifying with ATP7A isolated from BCS or copper-treated cells were not evidently different (*Figure 1D*, compare lanes 2 and 2'

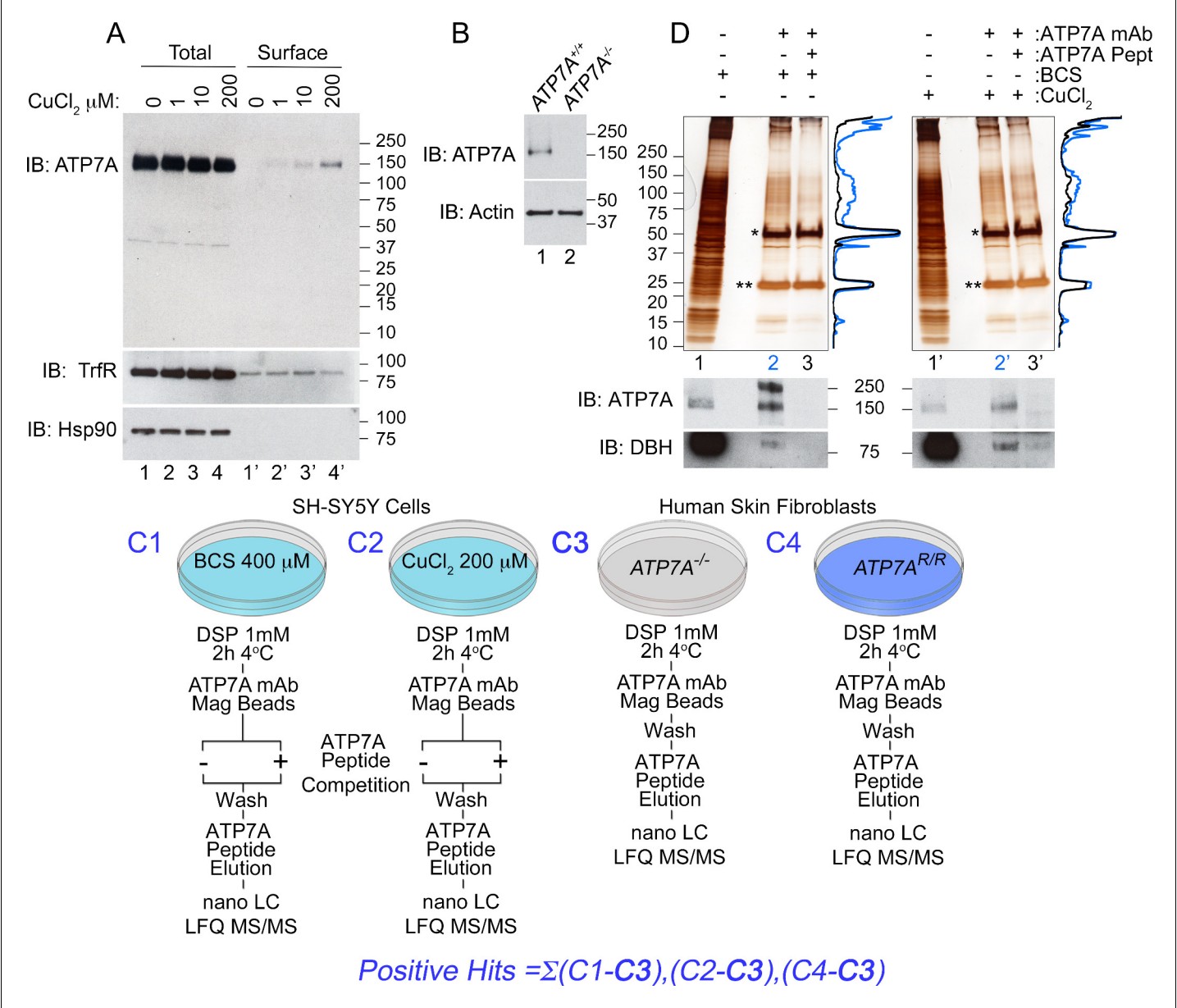

**Figure 1.** Isolation of ATP7A interactome. (**A**) In SH-SY5Y neuroblastoma cells, the addition of increasing amounts of copper leads to an increase of ATP7A at the cell surface as measured by surface biotinylation followed by streptavidin pull-down (lanes 1'–4'), while the total levels of ATP7A remain unchanged (lanes 1–4). Transferrin receptor shows consistent surface expression regardless of copper addition (lanes 1'–4'). The cytosolic chaperone Hsp90 was used to assess the selectivity of streptavidin pulldowns (**B**) The monoclonal ATP7A antibody used in these studies recognizes a single band by immunoblot. This band is missing in ATP7A null human Menkes fibroblasts (lane 2). (**C1–C4**) Experimental designs to isolate ATP7A interactomes. ATP7A immunoaffinity chromatography was performed in two cell types, SH-SY5Y cells (**C1-2**) and human skin fibroblasts (**C3-4**). In the former, left, SH-SY5Y cells were incubated with either 400 µM BCS (**C1**), a copper chelator, or 200 µM CuCl2 for 2 hr (**C2**). Cells were crosslinked with DSP, cell extracts were immunoprecipitated with the monoclonal ATP7A antibody either in the absence or presence of 22 µM ATP7A antigenic peptide. The same peptide was used to elute samples, which were then analyzed by label free quantitative mass spectrometry or silver stain (**D**). C3-4, ATP7A immunoaffinity chromatography was performed in ATP7A-null human skin fibroblasts (**C3**) as well as the same cells recombinantly expressing ATP7A (**C4**). The experiment was performed as in SH-SY5Y cells, with the exception of the ATP7A antigenic peptide being omitted for outcompetition. (**D**) Silver stain from ATP7A immunoprecipitation depicted in (**C**) except that immunocomplexes were eluted with Laemmli sample buffer. Immunoprecipitations were performed for BCS-treated (lanes 1–3) and CuCl2 treated (lanes 1'–3') SH-SY5Y cell extracts. Asterisks indicate immunoglobulin G chains, and densitometry profiles show differential protein enrichment in samples immunoprecipitated with (lanes 3 and 3', black traces) and without (lanes 2 and 2', blue traces) the antigenic ATP7A peptide. Below are immunoblots performed in parallel revealing positive identification of ATP7A and known interacting partner dopamine beta hydroxylase (DBH).

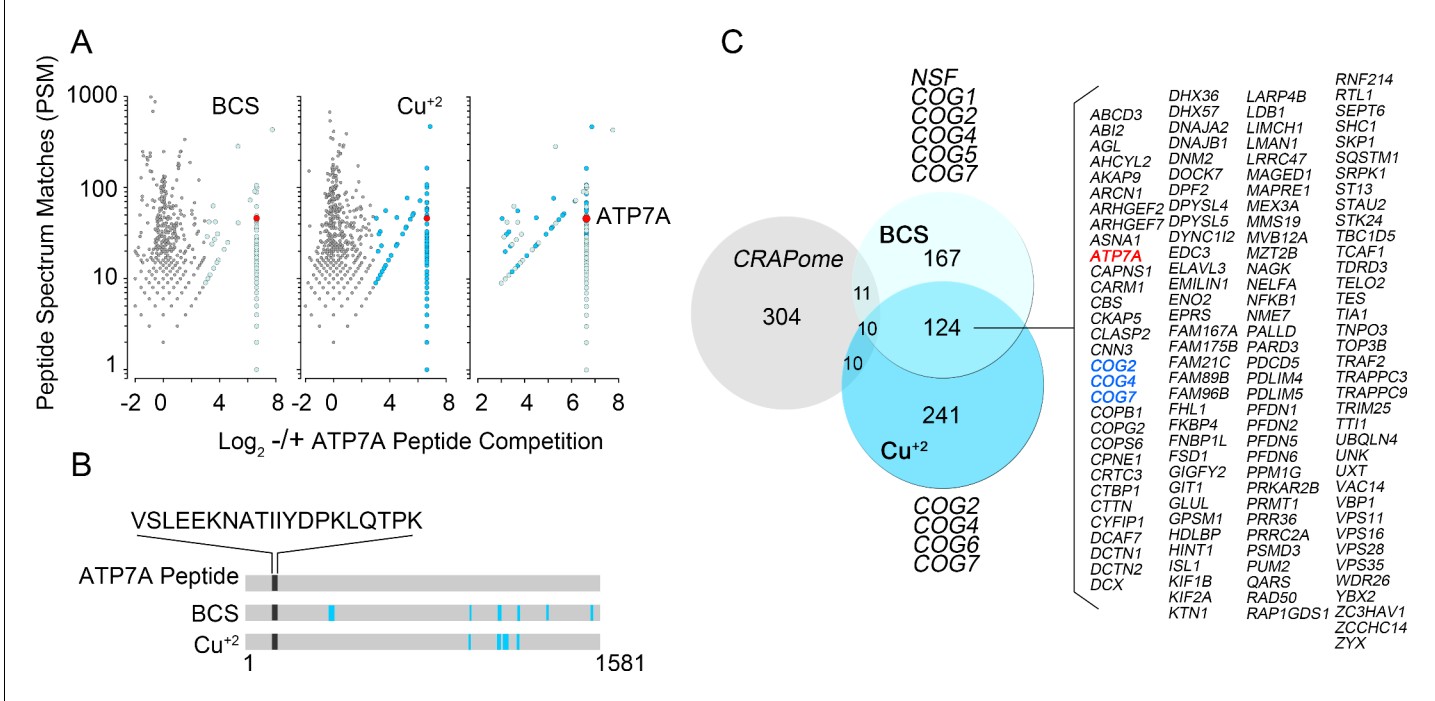

**Figure 2.** Identification of ATP7A interactome components by mass spectrometry. (**A**) Peptide spectrum matches (PSM) from quantitative mass spectrometry of proteins co-isolating with ATP7A are plotted for cells incubated with copper chelator BCS (left), $CuCl_2$ (middle), and peptides identified in both samples (right). Blue dots represent peptides that were enriched 2-fold over negative control samples incubated with an antigenic ATP7A peptide. (**B**) ATP7A peptides identified by mass spectrometry. Peptides corresponded to the antigenic peptide sequence shown above the black line as well as other ATP7A peptides identified via mass spectrometry (blue lines). (**C**) Five hundred and forty one proteins co-isolated with ATP7A, one hundred thirty four of which were present regardless of cellular copper status are listed. Three COG subunits were present in both BCS and copper-treated samples (blue text), the other three subunits were found either in BSC or copper samples. Curation with a dataset from the CRAPome reveals minimal overlap.

and traces, and *Figure 2C*). Therefore, we reasoned that proteins common to BCS and copper conditions would represent stable and/or strong interactors carried by ATP7A irrespective of its subcellular location (*Figure 2C*).

Quantitative mass spectrometry identified five hundred and forty one positive hits co-isolating with ATP7A, as defined in *Figure 1C* and *Supplementary files 1–3*. These five hundred and forty one proteins are the result of a curation with a dataset obtained under the same conditions but using cellular homogenates from ATP7A null human cells (*Figure 1C3*, see *Supplementary files 1–3*). We confirmed the quality of this curation step by comparing the five hundred and forty one positive hits to a published control dataset of three hundred and thirty five proteins that spuriously bind magnetic beads (*Figure 2C*, CRAPome) (*Mellacheruvu et al., 2013*). The overlap between our five hundred and forty one ATP7A interactors and the CRAPome was just thirty one proteins, a 5.7% overlap (*Figure 2C*, CRAPome, *Supplementary file 3*). One hundred and thirty four proteins co-isolated with ATP7A regardless of whether cells were incubated with BCS or copper (*Figure 2A and C*, see *Supplementary files 1–3*). Mass spectrometry identified ATP7A peptides corresponding primarily to the antigenic primary sequence at the C-terminal domain of ATP7A (*Figure 2B*). The interactome includes known ATP7A interactors such as dopamine beta hydroxylase (*Gokhale et al., 2015a*); subunits of the WASH complex, FAM21 and WASH1 (*Phillips-Krawczak et al., 2015*; *Ryder et al., 2013*); subunits of the retromer complex, VPS26 and 35 (*Phillips-Krawczak et al., 2015*; *Steinberg et al., 2013*); and components of clathrin-coated vesicles such as CLTB (*Holloway et al., 2013*; *Montpetit et al., 2008*; *Yi and Kaler, 2015*; *Hirst et al., 2012*). In order to prioritize candidate proteins for functional studies, we analyzed the ATP7A interactome with gene ontology algorithms (*Figure 3*, *Supplementary file 3A–B*). Gene ontological categories were enriched in Golgi-related terms such as Golgi transport complex (GO:0017119, p<5E-6), which

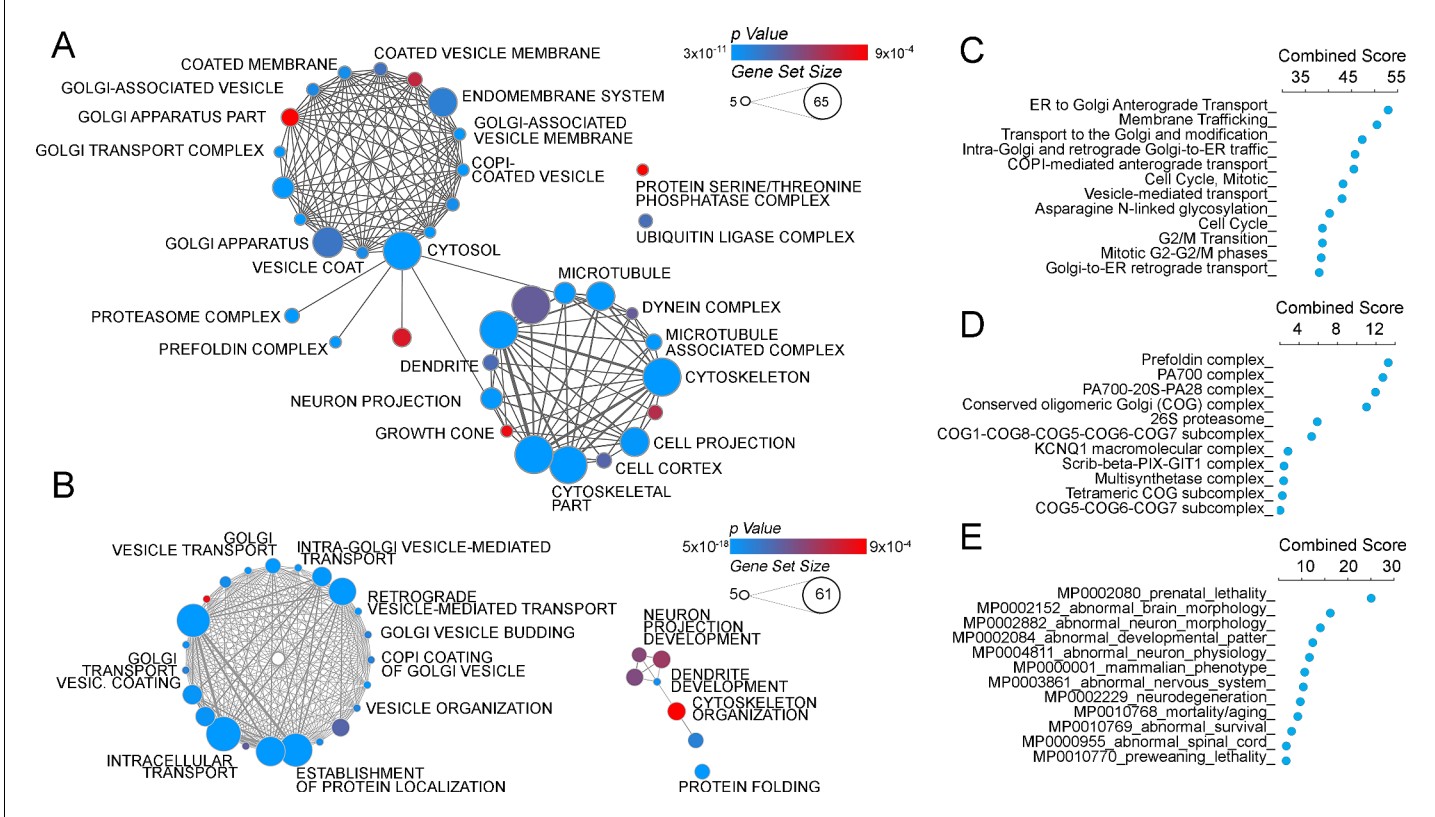

**Figure 3.** The ATP7A interactome enriches gene products implicated in Golgi function and neuropathologies. (**A–B**) The gene ontology algorithm DAVID was used to analyze the ATP7A interactome using the GO Terms Cellular Component (**A**) and Biological Process (**B**). Size of the circles increases with gene set size, and p-values are represented by colors ranging from blue (p=3×10$^{-11}$) to red (p=9×10$^{-4}$). Lines connecting circles depict ontology categories with shared gene products. (**C–F**) The ATP7A interactome was also evaluated using ENRICHR algorithm, to characterize dataset enrichment in GO Term Biological Process (**C**), protein complexes from the CORUM database (**D**), and phenotypes in mice (**E**). Significance is represented as a combined score (z-score x -log(p-value)).

contained six of the eight subunits of the Golgi localized COG tethering complex. Three of these six COG subunits were identified in the BSC and copper-challenged ATP7A interactomes (*Figure 2C*, COG2, 4 and 7). Additionally, bioinformatics identified neuronal ontological terms including growth cone, neuron projection, and dendrite (GO:004300, 0030426, 003042, respectively; all p<10E-4, *Figure 3A*, *Supplementary file 3A–B*). Similar bioinformatic results were obtained irrespective of the algorithm used, DAVID (*Figure 3B*, see *Supplementary file 3A*) or ENRICHR (*Figure 3C*, see *Supplementary file 3B*) (*Huang et al., 2007*, *2009*; *Chen et al., 2013*). The ATP7A interactome is overrepresented in biological processes related to membrane trafficking and vesicular transport, Golgi-related transport in particular. Proteins contained in these ontological terms were among the products identified in both BSC and copper-challenged ATP7A interactomes, such as the COG complex subunits (COG2, 4 and 7, *Figure 2C*). In fact, the COG complex was among the most prominently enriched complexes present in the ATP7A interactome (CORUM database, *Figure 3D*, combined score >7, see *Supplementary file 3B*) (*Ruepp et al., 2010*). These gene ontology tools point to the Golgi-dependent and neuronal trafficking mechanisms as key components of the ATP7A interactome.

We next asked what traits would associate with loss-of-function mutations in components of the ATP7A interactome using mouse phenotypic databases (*Chen et al., 2013*). Analysis of gene sets describing mouse phenotypes demonstrated that gene products contained in the ATP7A interactome were significantly enriched in categories describing neuronal pathology such as abnormal brain-neuronal morphology and neurodegeneration (*Figure 3E*, combined score >9.5, *Supplementary file 3B*). To further our understanding of phenotypes associated with members of

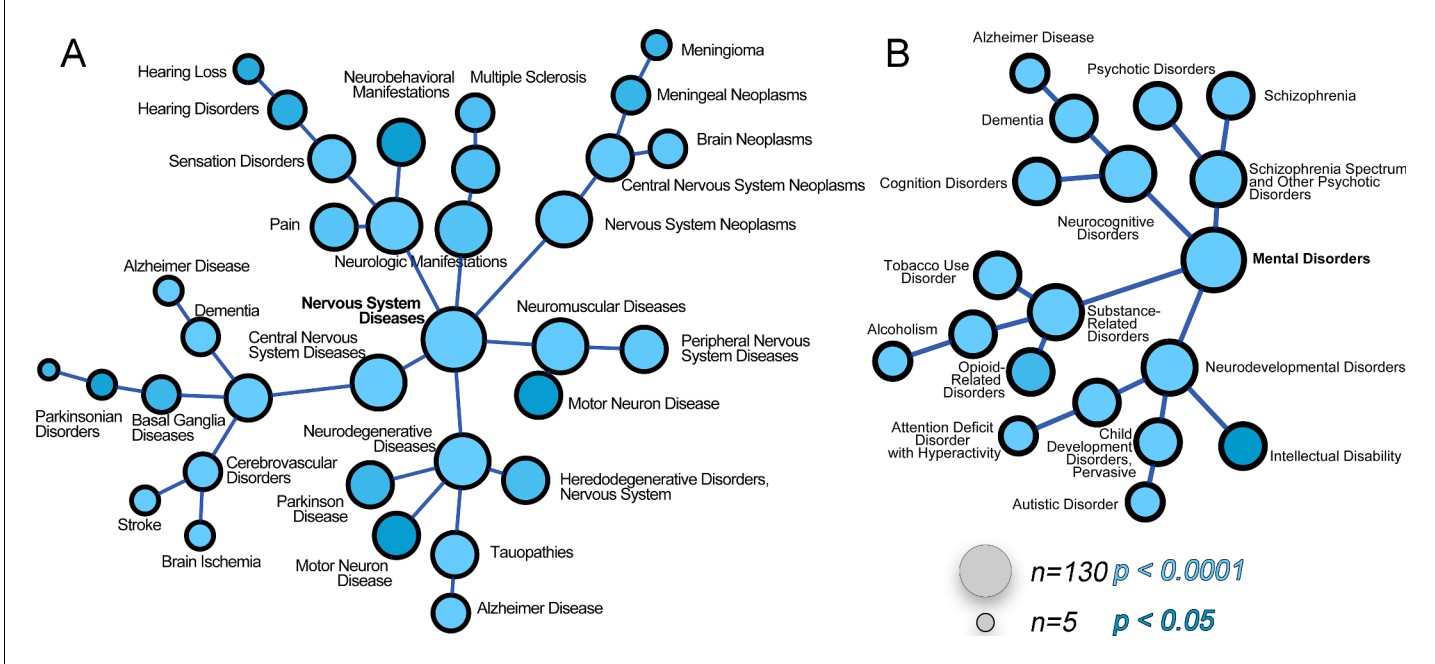

**Figure 4.** The ATP7A interactome enriches gene products associated with nervous system diseases and mental disorders. (A–B) The ATP7A interactome was analyzed using the GDA bioinformatics tool, which derives disease-gene links from human databases OMIM and Genopedia. Nervous system diseases (A) and mental disorders (B) were both significantly enriched in the dataset (p<0.0001). Circle size represents number of gene product from the ATP7A interactome in the disease category. Shades of blue depict p values.

the ATP7A interactome, we utilized the GDA bioinformatic tool that derives disease-gene links from human databases OMIM and Genopedia (*Park et al., 2014*). This analysis revealed a significant association with neurodegenerative nervous system diseases (*Figure 4A*, *Supplementary file 3C*) and psychiatric disorders (*Figure 4B*, *Supplementary file 3C*). The ATP7A interactome enriched 62 neurodegeneration associated or causative genes (MeSH, Medical Subject Heading C10.574, p<0.0001), which include COG2, VPS26, VPS35, DBH, and NSF. Moreover, 181 gene products contained in the ATP7A interactome were associated with mental disorders (Medical Subject Heading F03, p<0.0001, *Supplementary file 3C*). Within the mental disorder category the MeSH term neurodevelomental disorders stood out (Medical Subject Heading F03.625, p<0.0001, *Supplementary file 3C*) represented with 34 gene products. These analyses indicate that the ATP7A interactome enriches gene products previously implicated in neurodegenerative and neurodevelopmental disorders, suggesting that these associations could participate in the neuropathogenesis of ATP7A genetic deficiencies.

We prioritized for confirmation ATP7A interactome candidates due to their association with neurodegenerative and neurodevelopmental disorders. We immunoprecipitated ATP7A from cell extracts of cross-linked SH-SY5Y neuroblastoma cells and immunoblotted for proteins of interest. We focused on proteins that were either co-enriched with ATP7A to a similar extent as the bait (fold of enrichment log2 >6), were present in the BCS and copper treated ATP7A interactomes (*Figure 2C*), were associated with or causative of neurodegeneration (VAC14, strumpellin, NFKB1, and GIGYF2/PARK11) as well as mental disorders (NSF, DOCK7, GIGYF2, and COG complex subunits, *Figure 3*), and/or were present in compartments where ATP7A traffics to and from. The WASH complex subunit strumpellin and clathrin heavy chain (CHC) served as controls for previously described ATP7A interacting proteins (*Phillips-Krawczak et al., 2015*; *Ryder et al., 2013*; *Holloway et al., 2013*; *Montpetit et al., 2008*; *Yi and Kaler, 2015*; *Hirst et al., 2012*). The ATP7A monoclonal antibody co-immunoprecipitated strumpellin and clathrin heavy chain, and other proteins belonging to the ATP7A interactome listed above (*Figure 5A*, lane 2). The addition of ATP7A immunogenic peptide prevented the coprecipitation of ATP7A with these selected interactome components (*Figure 5A*, lane 3). To further establish specificity of our immunoprecipitation method,

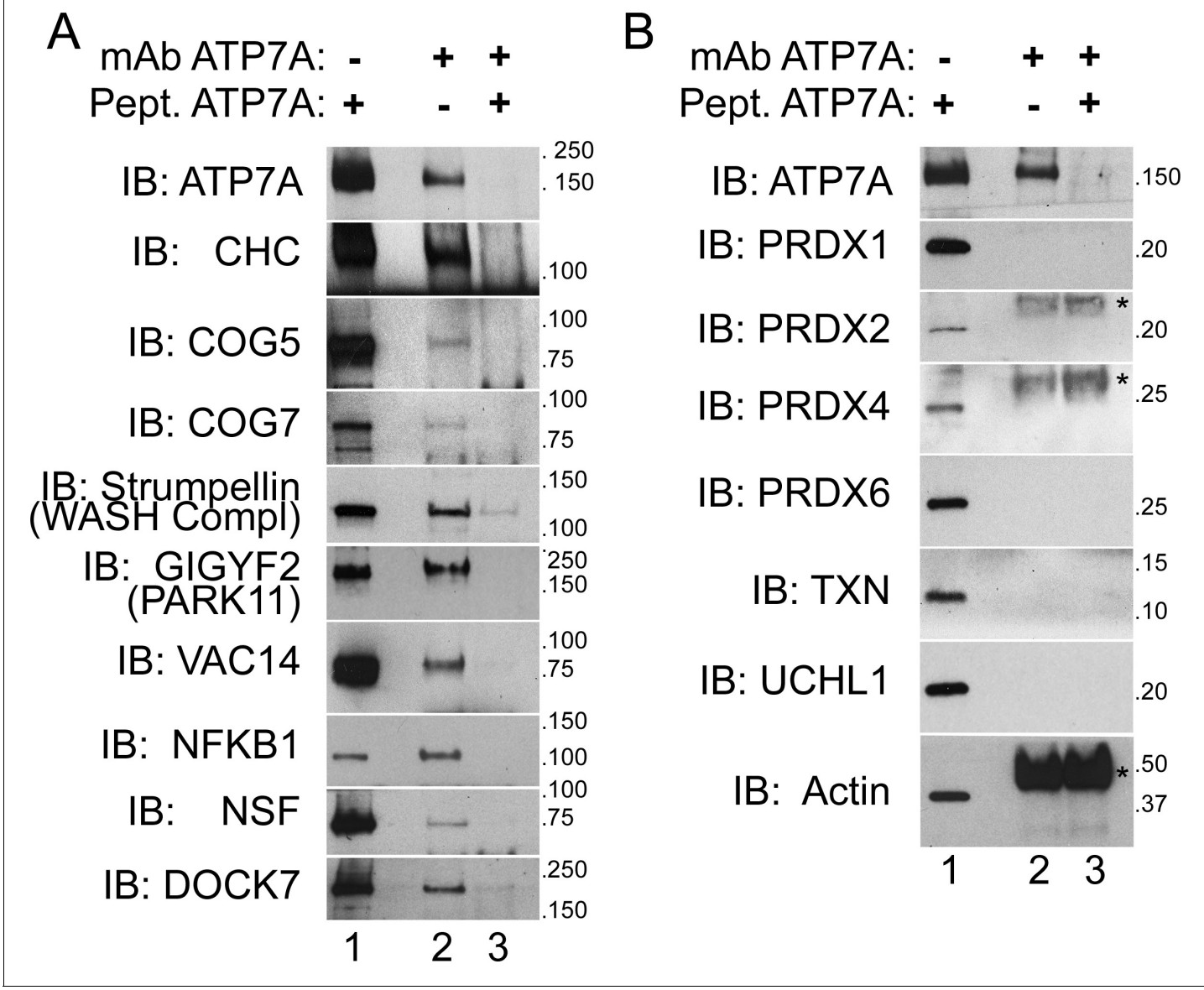

**Figure 5.** ATP7A co-immunoprecipitates with proteins implicated in neurodegeneration and neurodevelopmental disorders. (A–B) ATP7A was immunoprecipitated from DSP-crosslinked SH-SY5Y neuroblastoma cell lysates. Whole cell extracts (lanes 1) and immunoprecipitated samples (lanes 2 and 3) were resolved by SDS-PAGE and analyzed by immunoblot. Lane 3 contains samples in which an antigenic ATP7A peptide was added during immunoprecipitation as a negative control. (A) ATP7A and previously characterized interactors, clathrin heavy chain (CHC) and strumpellin, were selectively identified, along with newly identified interactome components that were highly enriched by mass spectrometry and associated with neuropathologies. Note the presence of two COG complex subunits. (B) Abundant cytosolic proteins that fell below the significant fold of enrichment by mass spectrometry or were identified in the negative control samples failed to co-purify with ATP7A. None of these proteins coprecipitate with ATP7A demonstrating the specificity of the interactions depicted in A. Asterisks denote mouse IgG chains.

we immunoblotted for abundant cytosolic proteins that were identified by mass spectrometry either in the negative control samples or that fell below the threshold of enrichment established for significance. These proteins include actin, several peroxiredoxins, thioredoxin, and the ubiquitin hydrolase UCHL1 (*Figure 5B*). None of these proteins co-isolated with ATP7A even in the presence of a cross-linker (*Figure 5B*, lane 2). Neither specific interactors of ATP7A (*Figure 5A*) nor proteins used as controls (*Figure 5B*) changed their association with ATP7A after pretreating cells with copper chelator (BSC-treated cells, *Figure 1C*) or copper (data not shown). These results validate the ATP7A

interactome and confirm that this interactome enriches neurodevelopmental and neurodegenerative gene products.

## The COG complex is necessary for copper transporter stability and cell surface expression

We selected the COG complex to test the hypothesis that genetic defects in components of the ATP7A interactome impair copper homeostasis by metal transporters for three reasons. First, we find that ATP7A co-purifies with six of the eight COG complex subunits to a similar extent (*Figure 2A and C*; *Figure 5A*). Secondly, our bioinformatics analysis prioritizes this complex among components of the ATP7A interactome (*Figure 3A–E*). Finally, mutations in the human COG subunits result in neurological defects some overlapping with Menkes disease (*Figures 3E* and *4*) (*Foulquier et al., 2006*; *Kodera et al., 2015*; *Reynders et al., 2009*; *Paesold-Burda et al., 2009*; *Fung et al., 2012*; *Rymen et al., 2012*; *Lübbehusen et al., 2010*; *Huybrechts et al., 2012*; *Shaheen et al., 2013*; *Wu et al., 2004*; *Morava et al., 2007*; *Zeevaert et al., 2009*; *Foulquier et al., 2007*)..

We used HEK293 cells rendered null for COG subunits 1 (COG1$^{\Delta/\Delta}$) or 8 (COG8$^{\Delta/\Delta}$) using CRISPR-Cas9 genome editing (*Blackburn and Lupashin, 2016*; *Bailey Blackburn et al., 2016*). These cells reproduce cellular phenotypes associated with COG complex deficiencies, such as defective Golgi-dependent glycosylation of membrane proteins and degradation of Golgi-localized proteins in lysosomal compartments (*Blackburn and Lupashin, 2016*; *Bailey Blackburn et al., 2016*; *Shestakova et al., 2006*). Both COG null cell lines exhibited decreased expression of a known COG-dependent and Golgi-localized membrane protein, GPP130 (*Figure 6A*). In addition, the GPP130 protein was found to migrate faster during SDS-PAGE, an indication of defective GPP130 glycosylation in COG null cells (*Figure 6A*) (*Sohda et al., 2010*; *Oka et al., 2004*; *Zolov and Lupashin, 2005*). Similarly, ATP7A expression was reduced by ~25–50% (*Figure 6A–B*), and ATP7A SDS-PAGE migration was increased in COG-null cell lines (*Figure 6A*).

We determined whether COG deficiency also affected the surface expression of ATP7A by surface biotinylation. Wild type, COG1$^{\Delta/\Delta}$, and COG8$^{\Delta/\Delta}$ HEK293 cells were incubated at 37°C for two hours in the absence and presence of 200 µM CuCl$_2$, followed by surface biotinylation at 4°C. Biotinylated proteins were precipitated with streptavidin beads and analyzed by immunoblot with antibodies against the copper transporters ATP7A and CTR1 (*Figure 6C–F*). The efficiency of cell surface biotinylation was at nearly two-fold higher in COG-null cells as compared to wild type cells (*Figure 6C* compare odd and even lanes), thus we normalized all transporter surface expression levels as a ratio of the surface content to the biotinylation efficiency. Normalized surface levels of ATP7A were decreased in COG1$^{\Delta/\Delta}$ HEK293 cells as compared to controls (*Figure 6C*, compare lanes 5–6, *Figure 6E*). Copper addition to wild type HEK293 cells modestly decreased the normalized surface levels of ATP7A. In contrast, normalized ATP7A surface levels in COG1$^{\Delta/\Delta}$ HEK293 cells did not change after copper (*Figure 6C*, compare lanes 5 and 7 and 6 and 8; *Figure 6E*). The mobilization of ATP7A from the surface in the presence of excess copper in HEK293 cells is in contrast with the ATP7A response in SH-SY5Y neuroblastoma cells (*Figure 1A*). These findings demonstrate that the total and cell surface levels of ATP7A are decreased in cells with genetic defects in the COG complex.

Our findings suggested that either the ATP7A trafficking to and from the plasma membrane in the presence of copper differ between wild type HEK293 and SH-SY5Y cells or, alternatively, HEK293 cells are poorly responsive to a copper challenge. We tested the latter hypothesis by asking if the normalized surface content of plasma membrane copper transporter CTR1 was sensitive to copper addition. CTR1 is the main plasma membrane transporter required for copper influx into cells (*Gupta and Lutsenko, 2009*; *Kuo et al., 2001*). We used the well-known copper-induced endocytosis of CTR1 as a tool to assess if HEK293 cells respond to copper addition (*Figure 6C*) (*Petris et al., 2003*; *Clifford et al., 2016*). The CTR1 antibody recognized monomeric and oligomeric species when CTR1 was enriched in cell surface biotinylated proteins, thus we could not assess CTR1 total cellular expression (*Figure 6C* compare inputs 1–2 to lanes 5–8). However, we found pronounced down-regulation of normalized cell surface CTR1 in COG1$^{\Delta/\Delta}$ HEK293 cells in basal conditions (*Figure 6C* compare lanes 5 and 6, and *Figure 6F*). Addition of copper to wild type and COG1$^{\Delta/\Delta}$ HEK293 cells further exacerbated the depletion of CTR1 at the surface (*Figure 6C* compare lanes 5 and 7, lanes 6 and 8, and *Figure 6F*), demonstrating that wild type and COG1$^{\Delta/\Delta}$ HEK293 cells are

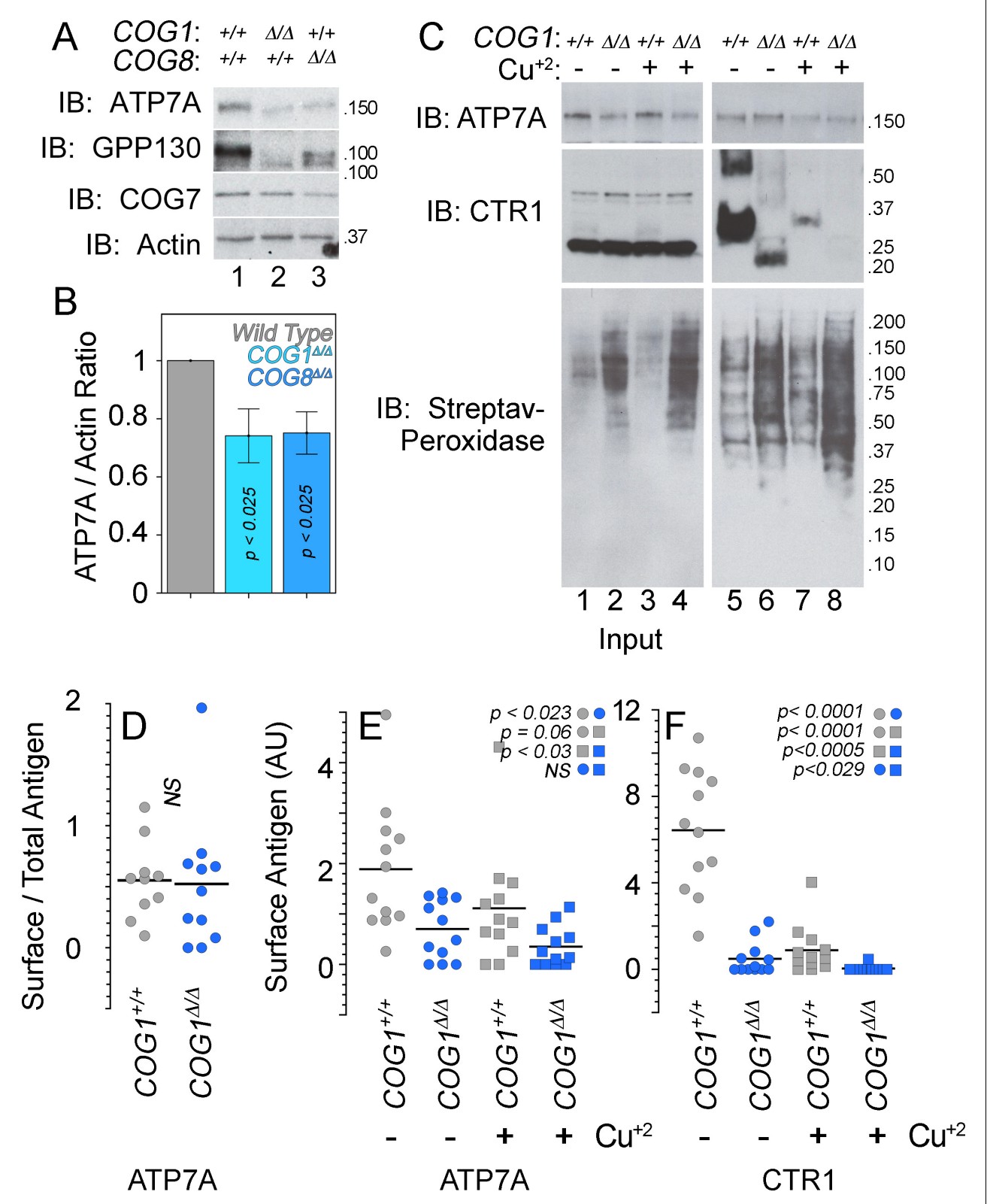

**Figure 6.** ATP7A stability and surface expression requires the COG complex. (**A–B**) Immunoblots for ATP7A, known COG-dependent protein GPP130, COG seven and actin were performed in wild type HEK293 cells (6A, lane 1) and cells null for COG subunit 1 (6A, lane 2, COG1$^{\Delta/\Delta}$) or COG subunit 8 (6A, lane 3, COG8$^{\Delta/\Delta}$). (**C**) Surface biotinylation of the same cell types was performed with (6C, even lanes) and without (6C, odd lanes) the addition of 200 μM CuCl$_2$ for two hours. Total protein extracts (6C, lanes 1–4) and surface biotinylated proteins precipitated by streptavidin beads (6C, lanes 5–8)

*Figure 6 continued on next page*

*Figure 6 continued*

were probed for ATP7A and CTR1, along with streptavidin-peroxidase to compare biotinylation efficiency. (**D–F**) ATP7A and CTR1 quantitations to measure the ratio of surface to total ATP7A (**D**), total surface ATP7A with and without copper (**E**), and total surface CTR1 with and without copper (**F**). D to F surface signals were corrected by the efficiency of biotinylation that was 1.84 ± 0.7 higher in COG1$^{\Delta/\Delta}$ cells (average ± SEM). Surface levels of ATP7A and CTR1 for each experimental condition were compared using non parametric Kriskal Wallis test followed by pairwise Mann-Whitney U test comparisons, n = 7.

responsive to copper challenge. These results demonstrate that COG complex genetic defects decrease the surface expression of two copper transporters, ATP7A and CTR1, thus suggesting complex copper metabolism phenotypes in COG deficiencies.

## COG complex genetic defects decrease cellular copper and modify the expression of ATP7A and other copper-sensitive transcripts

We addressed whether COG genetic defects impair cellular copper uptake by directly measuring the metal content in cells. In addition, we determined secondary effects of copper imbalances by quantifying transcripts whose expression is sensitive to metals and metal pathway dysfunction. We measured total copper and zinc content in wild type, COG1$^{\Delta/\Delta}$, and COG8$^{\Delta/\Delta}$ HEK293 cells with inductively-coupled plasma mass spectrometry. Copper was readily detectable in wild type HEK293 cells. However, copper content in COG-null HEK293 cells was below detection limit (1 ng/sample, *Figure 7A*). COG-null cellular copper phenotype was selective since zinc levels remained similar to wild type (*Figure 7B*). Addition of the copper-selective chelator BCS brought down copper content in wild type cells while addition of copper chloride or the copper ionophore disulfiram increased cellular copper levels in all three genotypes (*Figure 7A*). However, copper chloride uptake was impaired in COG null cells as compared to wild type HEK293 cells even when COG-null cells were exposed to 25 µM copper (*Figure 7A*). Only the addition of the copper ionophore disulfiram increased the copper content at or above wild type levels in COG null HEK293 cells (*Figure 7A*). These results demonstrate that the decreased expression of ATP7A and CTR1 observed in COG mutant cells results in selective copper deficiency.

Imbalances in cellular copper metabolism modify the expression of transcripts encoding copper transporters and metallochaperones that carry this metal to distinct molecules and subcellular compartments (*Robinson and Winge, 2010*). We previously used copper-sensitive gene expression to document copper metabolism imbalances in mutations of an ATP7A complex interactor, the BLOC-1 complex (*Gokhale et al., 2015a*). We predicted that transcripts of ATP7A and/or metallochaperones should be altered in COG deficient cells as a consequence of copper dyshomeostasis (*Figure 7A*). We focused on the metallochaperones ATOX1, which delivers copper to ATP7A (*Strausak et al., 2003*; *Voskoboinik et al., 1999*; *Walker et al., 2002*; *Lutsenko et al., 1997*); CCS, which carries copper to the mitochondrially localized SOD1 and that itself is imported into mitochondria (*Suzuki et al., 2013*; *Wang et al., 2013*); COX17, which is required for copper delivery to the mitochondrial cytochrome c oxidase (*Cobine et al., 2006*); and two isoforms of metallothioneins, both cysteine-rich proteins that bind metals in the cytoplasm (*Palmiter, 1998*).

We used a qRT-PCR assay capable of detecting changes in the expression of COG1 and COG8 transcripts in COG1$^{\Delta/\Delta}$ and COG8$^{\Delta/\Delta}$ HEK293 cells (*Figure 7C*). We normalized all transcript determinations against beta-actin mRNA. The housekeeping gene glyceraldehyde 3-phosphate dehydrogenase message showed no differences among cell genotypes when normalized to actin mRNA (*Figure 7C*, GAPDH). In stark contrast with the ATP7A protein expression levels, both COG1$^{\Delta/\Delta}$ and COG8$^{\Delta/\Delta}$ deficiency doubled ATP7A transcript levels as compared with wild type cells. These changes in ATP7A mRNA contrasted with the expression of the ATP7A metallochaperone, ATOX1, which remained unchanged (*Figure 7C*). Transcripts encoding metallothioneins IA and IIB were increased only in COG8$^{\Delta/\Delta}$ (MT1A and MT2A, *Figure 7C*). However, mRNA levels of two metallochaperones that traffic copper to mitochondria, CCS and COX17, were significantly down-regulated in both COG1$^{\Delta/\Delta}$ and COG8$^{\Delta/\Delta}$ cells (*Figure 7C*). CCS mRNA was reduced to 58% in both COG deficient cells whereas COX17 mRNA was decreased to 64% of the control values. These findings indicate that the expression of mitochondrial copper homeostasis factors is perturbed in COG deficient HEK293 cells. We further tested this hypothesis by measuring mRNA levels of mitochondrially

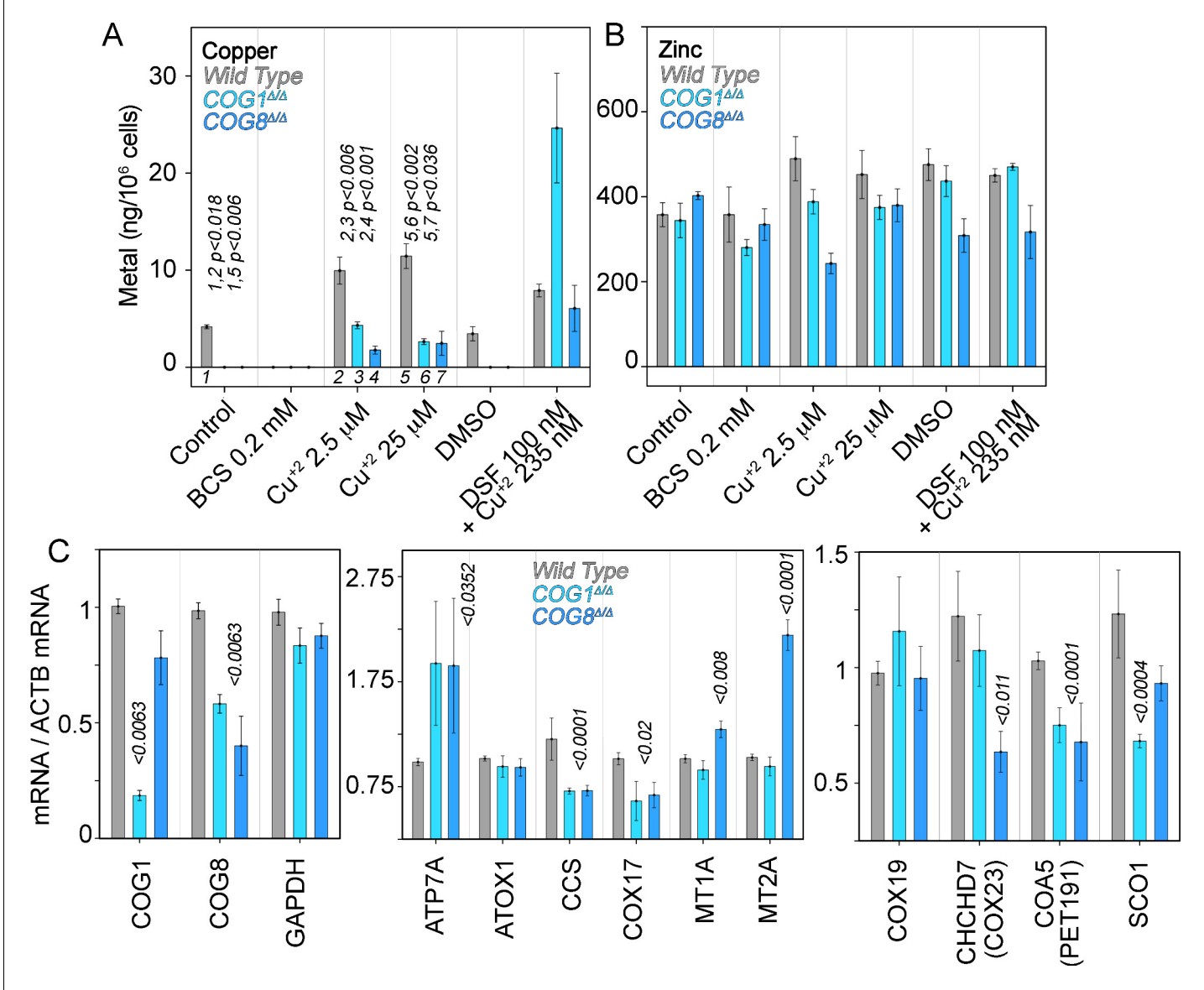

**Figure 7.** Copper content and expression of copper-sensitive transcripts is altered in COG deficient cells. Copper (**A**), zinc (**B**), and transcript (**C**) levels from wild type (grey bars), COG1$^{\Delta/\Delta}$ (light blue bars), and COG8$^{\Delta/\Delta}$ (dark blue bars) HEK293 cells were measured either by inductively coupled plasma mass spectrometry (**A-B**) or quantitative real-time PCR (**C**). In A and B cells were incubated for 24 hr with the indicated drugs in complete media with 10%FBS. Transcripts were normalized to beta-actin mRNA. Inductively coupled plasma mass spectrometry was performed in two independent biological replicates where each determination was in triplicate. Five independent biological replicates were performed for each determination in triplicate for quantitative real-time PCR. For metal determinations, significant p-values were determined by ANOVA followed by Dunnett test. p-values associated with transcript changes were determined by non-parametric Kriskal Wallis test followed by pairwise Mann-Whitney U test comparisons; all unlabeled comparisons are not significant in (**C**).

localized co-factors that work in concert with COX17 to load copper into cytochrome c oxidase (COX19, COX23, PET191, and SCO1, *Figure 7C*) (*Cobine et al., 2006*). Of these factors PET191/ COA5 mRNA was significantly decreased to ~70% in both COG1$^{\Delta/\Delta}$ and COG8$^{\Delta/\Delta}$ HEK293 cells (*Figure 7C*). In contrast, SCO1 and COX23 were selectively down-regulated in COG1$^{\Delta/\Delta}$ and COG8$^{\Delta/\Delta}$ cells, respectively (*Figure 7C*). Thus, COG complex deficient cells have altered expression of at least four transcripts implicated in the delivery of copper to mitochondrial enzymes. These

results suggest that COG complex deficiency cellular copper depletion leads to copper-dependent cellular and mitochondrial metabolism.

## COG-dependent ATP7A and CTR1 defects impairs copper homeostasis

We assessed the effects of increasing extracellular copper on cellular and mitochondrial metabolism with tetrazolium salts. Tetrazolium is reduced into formazan by NAD(P)H-dependent oxidoreductases and dehydrogenases localized to cytoplasm and mitochondria (*Berridge et al., 2005*). We measured 2-(4,5-dimethyl-2-thiazolyl)-3,5-diphenyl-2H-tetrazolium bromide reduction (MTT) in HEK293 cells either wild type, COG1$^{\Delta/\Delta}$, COG8$^{\Delta/\Delta}$, or carrying combined defects in COG1 and COG8 (*Figure 8*, COG1,8$^{\Delta/\Delta}$). Cells were exposed to increasing extracellular copper concentrations for 72 hr and MTT reduction was measured. Wild type cells exposed to low copper, 3–20 µM, increased MTT reduction as compared to wild type cells incubated with media alone (*Figure 8A*, grey symbols). In contrast, COG1$^{\Delta/\Delta}$, COG8$^{\Delta/\Delta}$, and COG1,8$^{\Delta/\Delta}$ HEK293 cells exposed to low copper failed to increase MTT metabolization (*Figure 8A*, blue symbols and *Figure 7A*). Irrespective of the cell genotype, all cells experienced a decrease of MTT metabolization at copper concentrations above 50 µM, likely due to copper toxicity (*Figure 8A*). These COG- and copper-dependent phenotypes do not reflect general cellular sensitivity to metal-based toxicants as assessed with cisplatin, a cytotoxic agent whose import into HEK293 cells does not require CTR1 copper transporter activity (*Figure 8B*) (*Bompiani et al., 2016*). These results demonstrate that COG deficient cells fail to respond to extracellular copper as predicted from the defects in ATP7A and CTR1 surface transport mechanisms.

We hypothesized that if decreased cellular copper and normalized surface levels of ATP7A and CTR1 in COG deficient HEK293 cells (*Figure 6*) prevent MTT metabolization, then direct copper delivery across the plasma membrane via a copper ionophore should revert MTT phenotypes (*Figure 8C–D*). Disulfiram is a cell permeant copper chelation agent that inhibits copper dependent enzymes in diverse compartments including mitochondria (*Simonian et al., 1992*; *Kuroda and Cuéllar, 1993*; *Goldstein, 1966*; *Gaval-Cruz and Weinshenker, 2009*). However, disulfiram complexed with copper increases metal cellular levels (*Cen et al., 2004*; *Allensworth et al., 2015*) (*Figure 7A*), and rescues respiration phenotypes in CTR1 null cells (*Schlecht et al., 2014*). We incubated wild type and COG deficient HEK293 cells with disulfiram in the absence (*Figure 8C and E*) or presence of copper (*Figure 8D and F*). Cells were incubated for 24 hr to minimize the effect of modifications in cell numbers on MTT activity readings. We controlled for cell numbers with a crystal violet colorimetric assay (*Feoktistova et al., 2016*) (*Figure 8E–F*). Addition of increasing disulfiram to wild type and COG-null HEK293 cells did not affect MTT activity at low concentrations, yet disulfiram above 25 nM decreased MTT activity (*Figure 8C*). None of these disulfiram concentrations significantly affected cell numbers (*Figure 8E*) indicating that MTT metabolization was impaired by the copper chelation activity of disulfiram at high doses. Next, we added increasing disulfiram concentrations to wild type and COG null HEK293 cells in the presence of 2.5 µM of copper (*Figure 8D and F*). Disulfiram concentrations above 25 nM in the presence of added copper decreased MTT activity and cell numbers irrespective of the cell genotype (*Figure 8D and F*). Thus, we focused on disulfiram and copper conditions that did not compromise cell numbers (*Figure 8F*). Loading cells with copper using low concentrations of disulfiram (<25 nM) significantly increased MTT metabolization in COG1$^{\Delta/\Delta}$ and COG8$^{\Delta/\Delta}$ HEK293 cells as compared to wild type controls (*Figure 8D*, compare gray and blue symbols). These results show that delivering copper with a copper ionophore to bypass copper transporter plasma membrane defects increases the metabolization of MTT in COG null cells.

## Genetic interactions between ATP7A and COG complex subunits in *Drosophila melanogaster*

COG null HEK293 cells have copper-dependent cellular and metabolic phenotypes that can be rescued with a copper-loaded ionophore, disulfiram. We focused on ATP7A overexpression because it cell-autonomously decreases cellular levels of copper due to ATP7A mistargeting to the cell surface (*Hwang et al., 2014*; *Lye et al., 2011*). Thus, we hypothesized that phenotypes induced by genetically increasing ATP7A expression in neurons should be modulated by loss-of-function mutations in the COG complex, which decreases ATP7A expression (*Figure 6A–B*).

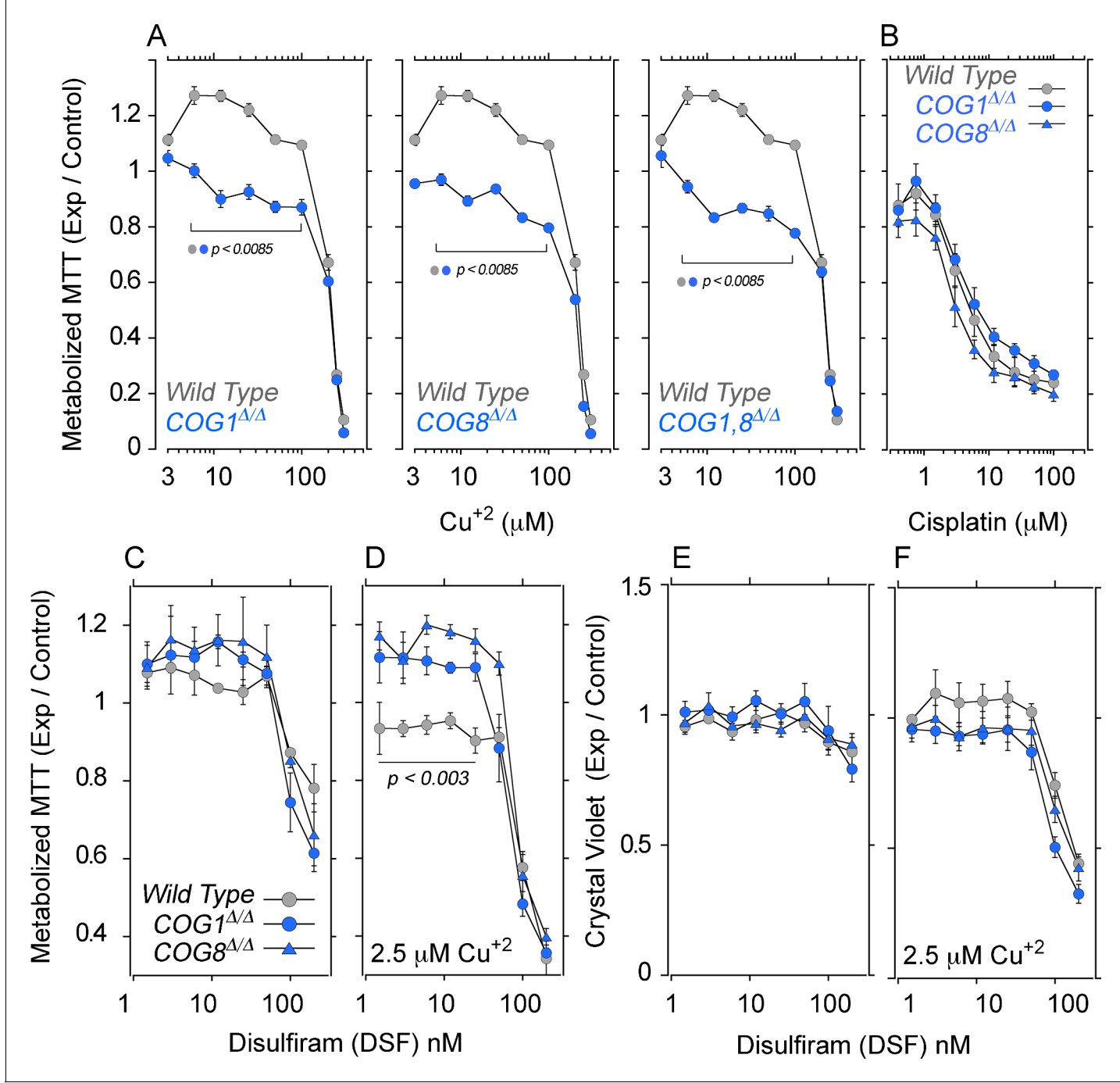

**Figure 8.** COG null cells possess copper metabolism defects. (**A**) Wild type, COG1$^{\Delta/\Delta}$, COG8$^{\Delta/\Delta}$ and COG1,8$^{\Delta/\Delta}$ HEK293 cells were incubated with CuCl$_2$ ranging from 3–300 μM for 72 hr. Each condition was carried out in quadruplicate, and the activity of NAD(P)H-dependent oxidoreductases was measured by the reduction of MTT. Each dot represents the average absorbance at 595 nm ± SEM, normalized to a baseline reading (n = 5). One Way ANOVA followed by Bonferroni's All Pairs Comparisons (**B**) Wild type and COG null cells were incubated with 0.4–100 μM cisplatin for 72 hr and MTT reduction was measured as above (n = 2). (**C–D**) The copper ionophore disulfiram (DSF) was added to wild type, COG1$^{\Delta/\Delta}$, and COG8$^{\Delta/\Delta}$ HEK293 cells for 24 hr either in concentrations ranging from 1.5–200 nM either in the absence (**C**) or presence (**D**) of 2.5 μM CuCl$_2$; each condition was carried out in quadruplicate. Reduction of MTT by NAD(P)H-dependent oxidoreductases was measured and normalized to a baseline reading with no drug added. Each dot represents the average of five independent biological replicates ± SEM. Non-parametric Kriskal Wallis test followed by pairwise Mann-Whitney U test comparisons. (**E–F**) Crystal violet staining was performed in parallel to MTT analysis to measure changes in cell number.

We first tested whether *Drosophila* ATP7A and COG complex subunits genetically interact to specify synapse morphology in the developing neuromuscular junction of the third instar larva (*Figure 9*). We overexpressed ATP7A in *Drosophila* neurons using the pan-neuronal *elav* GAL4 *c155* driver (C155) (*Lin and Goodman, 1994*). Overexpression of ATP7A reduced the cumulative synapse branch length; thus, inducing a collapse of the synapse as measured as an increased synaptic bouton density (*Figure 9A* image *C155>UAS-ATP7A*, *Figure 9B–C*, column 4. Compare 4 to control columns 1 and 2). In contrast, animals carrying one copy of the null allele *cog1[e02840]* increase cumulative synapse branch length while maintaining wild type synaptic bouton density (*Figure 9A–C*, column 3). As predicted by our hypothesis, overexpression of ATP7A in *cog1[e02840]* flies restored synaptic bouton density to wild type levels (*Figure 9A and B*, compare columns 4 and 5). These results demonstrate that a component of the ATP7A interactome, the COG complex, genetically interact with ATP7A to specify a neurodevelopmental synapse phenotype.

Second, we examined whether ATP7A and COG complex subunits genetically interact to specify neurodegeneration in the *Drosophila* adult nervous system (*Figure 10*). We controlled the expression of ATP7A in adult dopaminergic neurons, a group of cells frequently used to model Parkinson's disease in *Drosophila* (*Feany and Bender, 2000*; *Haass and Kahle, 2000*; *Li et al., 2000*; *Yang et al., 2003*; *Lin et al., 2010*). We drove the expression of UAS-ATP7A selectively in dopaminergic and serotoninergic neurons with the *dopa decarboxylase* (*Ddc*)-GAL4 driver (*Feany and Bender, 2000*). We reasoned that overexpression of ATP7A, which decreases cellular levels of copper (*Hwang et al., 2014*; *Lye et al., 2011*), should reduce the toxicity to copper diet exposure. We previously observed a high sensitivity to copper in the diet of wild type animals (*Gokhale et al., 2015a*). Copper feeding progressively increased mortality in wild type male (*Figure 10A*) and female adults (*Figure 10B*) over a period of three days. Overexpression of ATP7A in adult dopaminergic neurons was sufficient to significantly protect males and female adult animals from the toxic effect of copper feed at 48 hr (*Figure 10A–B*, (Ddc>UAS-ATP7A)). Similarly, mutation of the COG complex subunit *cog1* protected animals from copper diet induced death (*Figure 10A–B*, (Ddc x cog1[e02840])). In contrast, the mortality phenotype observed in animals overexpressing ATP7A was restored to the

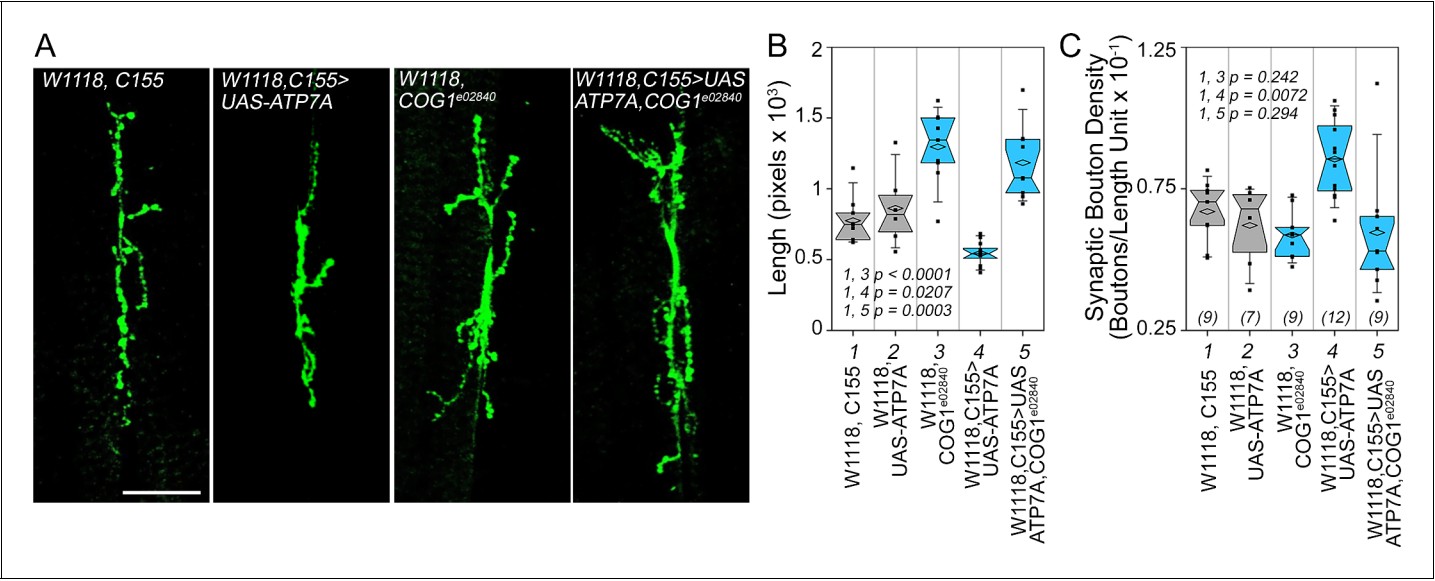

**Figure 9.** Drosophila ATP7A and COG1 genetically interact to specify *Drosophila melanogaster* synapse development. Third instar larvae neuromuscular junction synapses were stained with anti HRP antibodies (A) imaged and their morphology assessed using as parameters branch length (B) and bouton density (C). Scoring was done blind to the animal genotype. Control animals (C155 outcross, column 1; or UAS-ATP7A outcross, column 2), animals carrying one copy of the null allele *cog1[e02840]* (cog1 [e02840] outcrossed, column 3), flies overexpressing ATP7A in neuronal cells (c155>UAS-ATP7A; column 4), and animals overexpressing ATP7A and mutant for *cog1* (C155> UAS-ATP7A x *cog1[e02840]*, column 5) were analyzed. Numbers in parentheses and italics in C depict the number of animals. Statistical comparisons were performed with One Way ANOVA followed by Fisher's Least Significant Difference Comparison. Box plots depict percentiles fifth and 95th. Box line represents sample median and diamonds sample mean and notches mark the half-width.

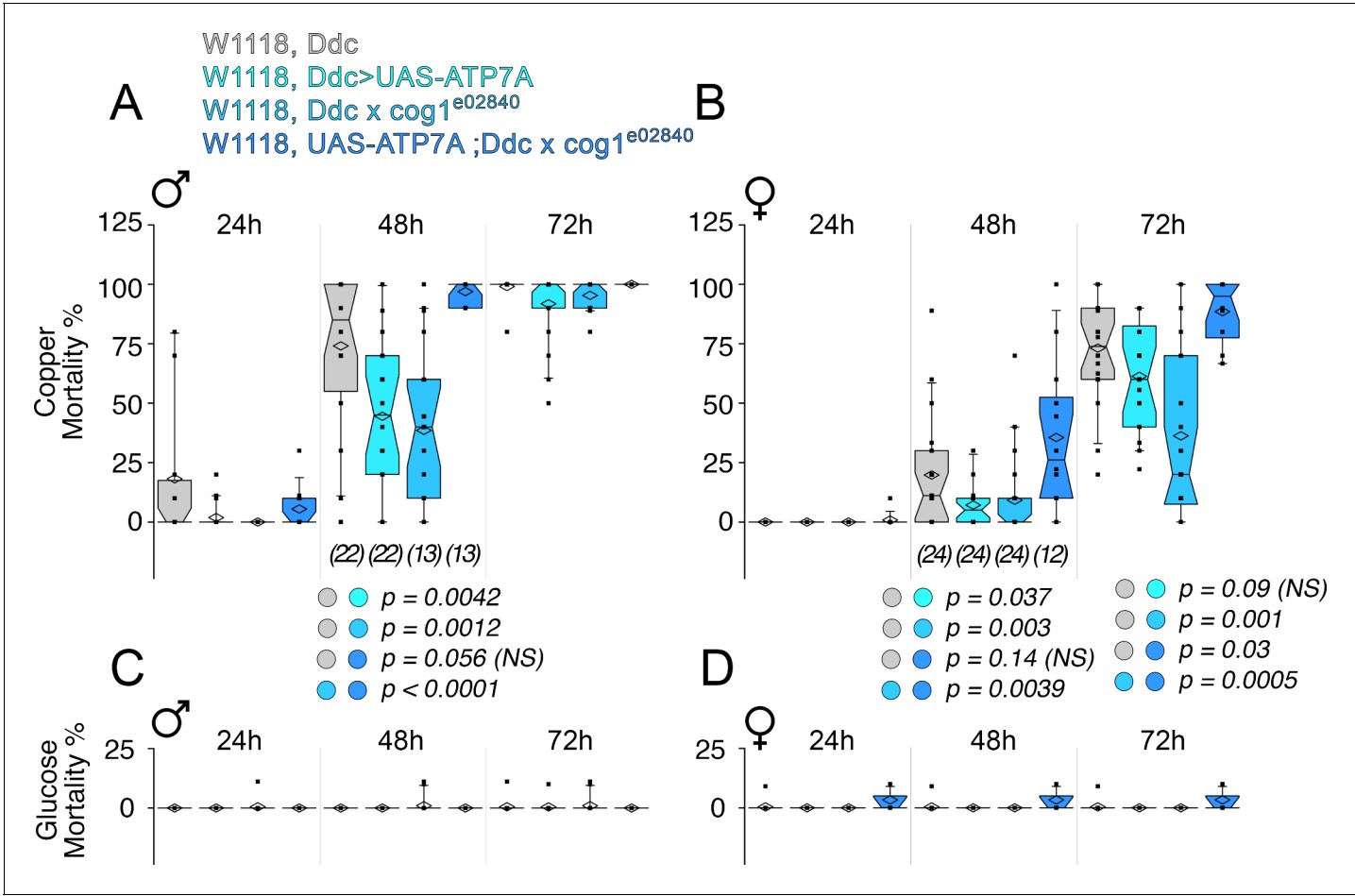

**Figure 10.** Drosophila ATP7A and COG1 genetically interact in dopaminergic neurons to specify copper-induced *Drosophila melanogaster* viability. Control animals (Ddc outcross), animals carrying one copy of the null allele *cog1e02840* (cog1 *e02840* outcrossed), flies overexpressing ATP7A in dopaminergic neuronal cells (Ddc>UAS-ATP7A), and animals overexpressing ATP7A and mutant for cog1 (UAS-ATP7A; Ddc x *cog1e02840*) were fed a glucose diet (**C-D**) or a glucose diet supplemented with 1 mM CuCl₂ for three consecutive days. Numbers in parentheses and italics depict the number of independent experiments each one performed with at least 10 animals per genotype. Statistical comparisons were performed with Non-parametric Kriskal Wallis test followed by pairwise Mann-Whitney U test comparisons. Box plots depict percentiles fifth and 95th. Box line represents sample median and diamonds sample mean and notches mark the half-width.

levels of wild type lethality by adding in trans a genetic defect in *cog1* (*Figure 10A–B*, (UAS-ATP7A; Ddc x cog1e02840)). Importantly, mortality was negligible when copper was omitted from the diet fed to animals of any genotype (*Figure 10C–D*). Our experiments demonstrate that the COG complex and ATP7A genetically interact in adult dopaminergic neurons to specify copper-dependent mortality.

## Discussion

We isolated and defined the ATP7A interactome to identify novel metal homeostasis mechanisms capable of modulating the expression of neurological traits. We selected candidate gene products from the ATP7A interactome based on three criteria; they strongly coenriched with ATP7A, were present in Golgi and post-Golgi compartments where ATP7A is present at steady state or traffics through after a copper challenge, and phenocopied some of the chief neurological phenotypes in Menkes disease. One such candidate that met all of these criteria, the COG complex, is associated with severe neuropathologies when mutated in humans and as we demonstrate have a severe depletion of cellular copper (*Foulquier et al., 2006*; *Kodera et al., 2015*; *Reynders et al., 2009*;

*Paesold-Burda et al., 2009*; *Fung et al., 2012*; *Rymen et al., 2012*; *Lübbehusen et al., 2010*; *Huybrechts et al., 2012*; *Shaheen et al., 2013*; *Wu et al., 2004*; *Morava et al., 2007*; *Zeevaert et al., 2009*; *Foulquier et al., 2007*). Here, we demonstrate that the COG complex is a hub where two copper transporters, ATP7A and CTR1, converge to regulate cellular copper content, homeostasis, synapse development and copper-induced neurodegeneration.

We compared the ATP7A interactome from cells treated in the presence of either excess copper or the copper chelator BCS (*Figure 1*). We reasoned that copper-dependent translocation of ATP7A from Golgi to post-Golgi compartments would reveal putative compartment-specific ATP7A interactors. However, we found that the most enriched proteins were associated with ATP7A regardless of whether cells were challenged with excess copper or not. It may be possible that we did not identify proteins that selectively associate with ATP7A in a copper-dependent manner because of our purification strategy. ATP7A peptides identified by mass spectrometry from the interactome were predominantly sequences enriched at the C-terminal domain of ATP7A, which could either be a reflection of the tertiary structure of the protein (*Figure 2*) or an indication that copper-induced protein associations with ATP7A mask the N-terminal domain from recognition by the antibody used as a bait. We favor the latter hypothesis because the antibody used to isolate the interactome is directed to the N-terminal domain, the same domain where the copper binding sites reside in ATP7A and a domain necessary for ATP7A interaction with ATOX1, the copper chaperone that shuttles copper to ATP7A (*Strausak et al., 2003*; *Voskoboinik et al., 1999*; *Walker et al., 2002*; *Lutsenko et al., 1997*). Consistent with this hypothesis, we did not detect ATOX1 in the ATP7A interactome although it is a robustly documented ATP7A interactor. However, our ATP7A antibody bait allowed the identification of both known and novel interacting proteins. ATP7A possesses 664 predicted and 13 experimentally tested interactors according to a comprehensive computational interactome of the human genome (*Garzón et al., 2016*). Our experimentally generated ATP7A interactome includes 541 proteins, of which only 21 proteins are in common with a computational ATP7A interactome. Therefore, the majority of the ATP7A interactome components identified here consisted of uncharacterized relationships. We identified several known trafficking proteins and complexes present at the plasma membrane, endosome, or Golgi complex that are known to regulate the trafficking of ATP7A, including the WASH complex and components of clathrin coated vesicles, as well as novel factors such as VAC14 and the COG complex (*Figure 2*). We independently confirmed that these trafficking components co-isolated with ATP7A (*Figure 5*). Strikingly, these trafficking proteins and complexes also associate with diverse neuropathologies, suggesting that they may share common mechanisms with the neuropathology observed in Menkes disease (*Figures 3–4*) (*Climer et al., 2015*; *Lenk et al., 2016*; *Bonifati et al., 2003*).

One prominent finding was the strong co-enrichment of COG complex subunits with ATP7A. We predicted that if the COG complex were responsible for trafficking of ATP7A, deletion of the complex would alter cellular copper content, subcellular localization and levels of copper transporters, the expression of copper-sensitive molecules, and cellular susceptibility to copper challenges. We found that in COG $^{\Delta/\Delta}$ cells, the total cellular and normalized surface levels of ATP7A are decreased (*Figure 6*), a molecular defect leading to selective copper depletion in COG null cells (*Figure 7A*). While the non-normalized surface levels of ATP7A measured by immunoblot were comparable in wild type and COG$^{\Delta/\Delta}$ cells at steady state, we normalized by the biotinylation efficiency, which was greater in the COG null cells. We attribute this increased biotinylation efficiency to a reduction in the surface glycocalyx negative charge in COG null cells, which would increase surface biotinylation efficiency with the anionic sulfo-NHS-biotin reagent (*Pokrovskaya et al., 2011*). When observing normalized surface levels of ATP7A after a copper challenge, we were surprised to find that, unlike in previously observed cell types including Caco-2, CHO cells, and human fibroblasts, ATP7A did not increase at the surface of wild type HEK cells (*Nyasae et al., 2007*; *Petris and Mercer, 1999*). We were unable to assess if this was due to an accelerated rate of endocytosis of ATP7A after copper addition, or a small pool of ATP7A at the Golgi unable to sustain increased ATP7A surface expression. However, we were able to document copper dependent membrane traffic of CTR1 in the same cells (*Gupta and Lutsenko, 2009*; *Kuo et al., 2001*; *Petris et al., 2003*; *Clifford et al., 2016*), thus excluding unresponsiveness to copper in HEK293 cells. We also assessed the total and normalized surface levels of CTR1, a copper importer that is known to internalize in the presence of excess copper. CTR1 normalized surface levels were dramatically decreased in COG$^{\Delta/\Delta}$ cells in resting conditions, and this deficiency was exacerbated by the addition of copper, leaving no

measurable CTR1 protein on the surface. This led us to hypothesize that COG deletion is perturbing not only copper regulation by ATP7A, but copper homeostasis in other cellular compartments that require the entry of copper from the plasma membrane via CTR1.

We decided to use copper-sensitive metabolism of MTT to determine whether the effects of COG deletion mutations affecting ATP7A and CTR1 surface expression extended beyond ATP7A-dependent Golgi trafficked enzymes, such as DBH, LOX and PAM (*Kaler, 2011*; *Lutsenko et al., 2007*). Cytochrome c oxidase, though not directly downstream of ATP7A, requires copper as a cofactor, and thus mitochondrial function is impaired by decreased activity of cytochrome c oxidase when copper delivery to mitochondria is impaired (*Leary et al., 2004*, *2007*). Our hypothesis that copper homeostasis is impaired in COG$^{\Delta/\Delta}$ cells is supported by reports describing that RNAi or mutation of COG subunits alters copper content in mammalian cells and yeast, respectively (*Schlecht et al., 2014*; *Malinouski et al., 2014*) and by our data that levels of copper and copper sensitive transcripts encoding proteins present in or transferring copper to different subcellular compartments are altered in COG null cells (*Figure 7*). Copper regulatory genes are known to undergo coordinated regulation in response to changing copper levels, and we found that multiple copper sensitive transcripts are decreased in COG$^{\Delta/\Delta}$ cells (*Barresi et al., 2016*). Of particular interest are two metallochaperones that deliver copper to mitochondria, CCS and COX17, whose message levels are reduced in COG null cells (*Robinson and Winge, 2010*; *Cobine et al., 2006*). We found that MTT metabolism is impaired in COG null cells due to defects in copper cellular homeostasis. These data suggest the existence of a mechanistic link between a Golgi-dependent metal buffering and copper-dependent oxidoreductases localized outside the Golgi complex. Several lines of evidence further support a model of a multicompartment regulation of copper homeostasis. Genetic evidence indicates that the expression of CTR1 is modulated by the mitochondrial metallochaperone SCO1 (*Hlynialuk et al., 2015*), down-regulation of mitochondrial components including complex I subunits (NDUFB4, NDUFA1) and the mitochondrial transporter SLC25A41 lead to global cellular changes in copper (*Malinouski et al., 2014*), and the elimination of ATP7A rescues mitochondrial metallochaperone mutant phenotypes (*Malinouski et al., 2014*; *Leary et al., 2013*).

Along with the neurodegeneration characteristic of Menkes disease, disruptions in copper homeostasis have been implicated in several other prevalent neurodegenerative diseases, including Parkinson's and Alzheimer's disease (*Greenough et al., 2016*; *Davies et al., 2016*). There is evidence that therapeutic modulation of copper levels may be effective in alleviating symptoms of or delaying the onset of Parkinson's and Alzheimer's disease, yet there are gaps in our understanding of the regulation of copper homeostasis, particularly in the brain (*Sparks and Schreurs, 2003*; *Brewer, 2015*; *Rose et al., 2011*). Polymorphisms in genes encoding the COG complex subunits COG4, 6, and 8 associate with neurodegenerative and neurobehavioral phenotypes in humans (*Li et al., 2008*; *Scharf et al., 2013*; *Kendler et al., 2011*; *Li et al., 2012*). Our bioinformatic studies also indicate that the COG2 subunit is one of 42 gene products present in the ATP7A interactome associated with tauopathies and Alzheimer's disease; one of fourteen genes products associated with Parkinson's disease; and one of the 49 gene products in the ATP7A interactome associated with neurocognitive disorders, such as dementia (*Figure 3*). Deletions of COG complex subunits lead to type II congenital disorders of glycosylation (CDG II) and are strongly associated with cerebral atrophy, developmental delay, hypotonia, ataxia and epilepsy (*Foulquier et al., 2006*; *Kodera et al., 2015*; *Reynders et al., 2009*; *Paesold-Burda et al., 2009*; *Fung et al., 2012*; *Rymen et al., 2012*; *Lübbehusen et al., 2010*; *Huybrechts et al., 2012*; *Shaheen et al., 2013*; *Wu et al., 2004*; *Morava et al., 2007*; *Zeevaert et al., 2009*; *Foulquier et al., 2007*). Since COG mutations drastically decreased the cellular copper levels (*Figure 7A*), it is possible that COG neurological phenotypes reflect in part decreased neuronal copper much like is the case in Menkes disease. Mutations in *cog1* prevent toxicity of dietary copper in Drosophila (*Figure 10*) supporting this concept of decrease copper content in COG deficient neurons. Similarities between Menkes and congenital disorders of glycosylation are suggested by neurological phenotypic overlap between these disorders. Common phenotypes include developmental delay, seizures, hypotonia and cerebral atrophy (*Kaler, 2011*; *Menkes, 1999*; *Menkes et al., 1962*; *Kennerson et al., 2010*; *Tümer, 2013*; *Kaler et al., 1994*). We genetically tested whether phenotypes caused by ATP7A dosage increase could be modulated by genetic defects in COG using the *Drosophila* developing neuromuscular synapse and adult dopaminergic neuron (*Figures 9–10*). Phenotypes due to overexpression of ATP7A in developing synapses and adult neurons could be reverted to wild type

by adding a mutation in *cog1*. Inspired by the effects of COG deficiency in human ATP7A (*Figure 6A–B*), we interpret this *Drosophila* rescue results as a reduction in the levels of overexpressed ATP7A in fly neurons due to a ATP7A down-regulation effect of the *cog1* mutant allele. This interpretation assumes cell autonomous effects of the *cog1* mutation on neuronal overexpressed ATP7A, and it does not consider possible non-cell autonomous contributions of the *cog1* genomic defect. Our findings suggest that part of the neurological phenotypes in COG genetic defects may be due to impaired copper content and metabolism as we report here.

While we initially focused on the role of the COG complex in ATP7A trafficking, it is clear that copper-related phenotypes downstream of COG extend beyond cuproenzymes that traverse the Golgi complex. Our strategy to identify an ATP7A interactome has revealed novel ATP7A interacting partners and may be a fruitful way to further explore networks related to other copper regulatory proteins. Our finding that the COG complex, a Golgi localized tether is implicated in mitochondrial copper homeostasis indicates that essential mineral homeostasis, such as copper, results from coordinated multicompartment metabolite sensing and response mechanisms.

## Materials and methods

### Cell culture, antibodies and primers

SH-SY5Y (ATCC Cat# CRL-2266, RRID:CVCL_0019) and HEK293T (ATCC Cat# CRL-3216, RRID: CVCL_0063) cells were cultured in Dulbecco's modified Eagle's medium (DMEM) supplemented with 10% fetal bovine serum (FBS) and 100 µg/ml penicillin and streptomycin (Hyclone) at 37°C in 10% $CO_2$. HEK293T cells deficient for COG subunits were generated as described in (*Blackburn and Lupashin, 2016*). Two lines of Menkes deficient fibroblasts were used: one in conjunction with a rescue line expressing recombinant ATP7A (described in *La Fontaine et al., 1998*) and the other in conjunction with a familial control (Coriell GM01981 and GM01983). The former was cultured in DMEM supplemented with 10% FBS and 100 µg/ml penicillin and streptomycin at 37°C in 5% $CO_2$, while the latter were cultured in minimum essential media (MEM) (Thermo Fisher 11095080) supplemented with 15% FBS and 100 µg/ml penicillin and streptomycin at 37°C in 5% $CO_2$. Antibodies and primers can be found in *Supplementary file 1*. Cell were tested for mycoplasma infection using the MycoAlert Mycoplasma Detection Kit from Lonza. None of the cell lines used in this study belong to the Database of Cross-Contaminated or Misidentified Cell Lines defined by the International Cell Line Authentication Committee.

### Immunoprecipitation of ATP7A

To assess interactions of ATP7A in the presence and absence of copper, we performed cross-linking in intact cells with dithiobis(succinimidylpropionate) (DSP) followed by immunoprecipitation as previously described but with the following modifications (*Gokhale et al., 2015a*). Briefly, SH-SY5Y neuroblastoma cells or ATP7A$^{-/-}$ and ATP7A$^{R/R}$ human fibroblasts were washed with ice-cold PBS with MgCl2 and CaCl2 (PBS/Mg/Ca). 200 µM Copper chloride or 400 µM BCS diluted in PBS/Mg/Ca buffer was then added to the respective plates and they were placed back in the 37°C incubator for 2 hr. The plates were then placed on ice, rinsed twice with PBS/Mg/Ca, and incubated with 10 mM DSP (Thermo Scientific 22585), diluted in PBS for 2 hr on ice. Tris, pH 7.4, was added to the cells for 15 min to quench the DSP reaction. The cells were then rinsed twice with PBS and lysed in buffer A (150 mM NaCl, 10 mM HEPES, 1 mM EGTA, and 0.1 mM MgCl2, pH 7.4) with 0.5% Triton X-100 and Complete anti-protease (catalog #11245200, Roche), followed by incubation for 30 min on ice. Cells were scraped from the dish, and cell homogenates were centrifuged at 16,100 × *g* for 10 min. The clarified supernatant was recovered, and at least 500 µg of protein extract was applied to 30 µl Dynal magnetic beads (catalog #110.31, Invitrogen) coated with 5 µl ATP7A antibody (NeuroMab Cat No. 75–142, RRID:AB_10672736), and incubated for 2 hr at 4°C. In some cases, immunoprecipitations were done in the presence of the antigenic ATP7A peptide as a control. The peptide (sequence VSLEEKNATIIYDPKLQTPK, custom made by Biosynthesis) was prepared in 10 mM MOPS and used at 22 µM. The beads were then washed 4–6 times with buffer A with 0.1% Triton X-100. Proteins were eluted from the beads with sample buffer. Samples were resolved by SDS-PAGE and contents analyzed by immunoblot or silver stain. In the case of the large-scale proteomic analysis, proteins eluted from the beads were combined and concentrated by TCA precipitation.

## Mass spectrometry

### Sample digestion

The IP beads were spun down and residual buffer was removed. Digestion buffer (200 ul of 50 mM NH4HCO3) was added and the bead solution was then treated with 1 mM dithiothreitol (DTT) at 25°C for 30 min, followed by 5 mM iodoacetimide (IAA) at 25°C for 30 min in the dark. Proteins were digested with 1 µg of lysyl endopeptidase (Wako) at room temperature for 2 hr and further digested overnight with 1:50 (w/w) trypsin (Promega) at room temperature. Resulting peptides were desalted with a Sep-Pak C18 column (Waters) and dried under vacuum.

### LC-MS/MS analysis

The dried peptides were resuspended in 10 µL of loading buffer (0.1% formic acid, 0.03% trifluoro-acetic acid, 1% acetonitrile). Peptide mixtures (2 µL) were separated on a self-packed C18 (1.9 µm Dr. Maisch, Germany) fused silica column (25 cm x 75 µM internal diameter (ID); New Objective, Woburn, MA) by a Dionex Ultimate 3000 RSLCNano and monitored on a Fusion mass spectrometer (ThermoFisher Scientific, San Jose, CA). Elution was performed over a 120 min gradient at a rate of 350 nl/min with buffer B ranging from 3% to 80% (buffer A: 0.1% formic acid in water, buffer B: 0.1% formic in acetonitrile). The mass spectrometer cycle was programmed to collect at the top speed for 3 s cycles. The MS scans (400–1600 m/z range, 200,000 AGC, 50 ms maximum ion time) were collected at a resolution of 120,000 at m/z 200 in profile mode and the HCD MS/MS spectra (0.7 m/z isolation width, 30% collision energy, 10,000 AGC target, 35 ms maximum ion time) were detected in the ion trap. Dynamic exclusion was set to exclude previous sequenced precursor ions for 20 s within a 10 ppm window. Precursor ions with +1, and +8 or higher charge states were excluded from sequencing.

### Database search

Spectra were searched using Proteome Discoverer 2.0 against human Uniprot database (90,300 target sequences). Searching parameters included fully tryptic restriction and a parent ion mass tolerance (±20 ppm). Methionine oxidation (+15.99492 Da), asaparagine and glutamine deamidation (+0.98402 Da) and protein N-terminal acetylation (+42.03670) were variable modifications (up to three allowed per peptide); cysteine was assigned a fixed carbamidomethyl modification (+57.021465 Da). Percolator was used to filter the peptide spectrum matches to a false discovery rate of 1%. Peptide spectral match (PSM) counts were used as the semi-quantitative measure.

## Inductively-coupled plasma mass spectrometry

Cells were washed three times with cold phosphate-based saline and detached by gentle squirting. Cell pellets were heated in a continuous cycle from 35°C to 95°C for 30 min with 2% nitric acid/$H_2O_2$. Copper and zinc were analyzed using inductively-coupled plasma mass spectrometry. Samples were quantified with an 8-point calibration using indium as an internal standard. The limits of detection were 1 ng/sample for copper and 0.1 ng/sample for zinc.

## Bioinformatic analysis

Gene list to disease associations were performed with GDA algorithm (http://gda.cs.tufts.edu/; *Park et al., 2014*). We performed gene ontology analysis with ENRICH (http://amp.pharm.mssm.edu/Enrichr/) (*Chen et al., 2013*), and Database Annotation, Visualization and Integrated Discovery (DAVID, https://david.ncifcrf.gov) (*Huang et al., 2007*, *2009*). Cytoscape with Enrichment Map plugin for visualizing DAVID outputs was used in order to depict integrations between GO terms associated with the ATP7A interactome as described (*Shannon et al., 2003*; *Merico et al., 2010*). Charts were narrowed down in order to simplify Cytoscape representations by eliminating broad GO terms. All Bioinformatic analyses are presented in *Supplementary file 3A—B*

## MTT assay

Cells were collected and seeded in a 96-wells plate at a density of $10 \times 10^3$ cells/well in DMEM +10% FBS+100 U/ml penicillin, 100 µg/ml streptomycin and incubated overnight. On day 2, cells were treated with serial dilutions of the appropriate drug (0.4–100 µM cisplatin, 3–300 µM copper chloride, or 1.5–200 nM disulfiram). After incubation at 37°C for either 24 or 72 hr depending on

the experiment, 20 µl 5 mg/ml (3-(4,5-Dimethylthiazol-2-yl)-2,5-Diphenyltetrazolium Bromide) (MTT) (Life Technologies M6496) was added to each well, and the plates were incubated for an additional 3.5 hr at 37°C. MTT was aspirated, 150 µl DMSO was added and cells were agitated on an orbital shaker for 15 min. The absorbance was read at 595 nm using a microplate reader. Each condition was carried out in quadruplicate, and MTT absorbance was expressed as percentage absorbance of untreated cells.

## Crystal violet staining

Cells were plated and treated with the appropriate drug as described for MTT experiments above. After incubation with drug, cells were washed once with PBS-Ca-Mg. Cells were then fixed for five minutes in 65% MeOH, followed by a fifteen minute fixation with 100% MeOH. The methanol was removed and wells were allowed to dry completely, after which 100 µl 0.1% w/v crystal violet in $H_2O$ was added to each well for five minutes. Plate was washed 3–5 times with $H_2O$, and crystal violet was solubilized with 2% sodium deoxycholate for 10 min. The absorbance was read at 595 nm using a microplate reader. Each condition was carried out in quadruplicate, and crystal violet absorbance was expressed as percentage absorbance of untreated cells (*Feoktistova et al., 2016*).

## Surface labeling and streptavidin pulldowns

Plates of ~75% confluent HEK293T cells were incubated for 2 hr at 37°C in the absence or presence of 200 µM copper chloride and then moved to an ice bath. Plates were then washed two times with ice-cold PBS-Ca-Mg (0.1 mM CaCl2 and 1 mM MgCl2 in PBS). All solutions used were ice-cold when applied to cells, and cells were maintained on an ice bath throughout. A biotin labeling solution of 0.5 mg/ml Sulfo-NHS-Biotin (Thermo Scientific 21217) in PBS-Ca-Mg was applied for 15 min, aspirated, and fresh labeling solution was applied for an additional 15 min. The labeling solution was then aspirated and cells were washed three times with 1 mg/mL glycine in PBS-Ca-Mg followed by one wash with plain PBS. Cells were collected from plates and incubated for 30 min in buffer A with 0.5% Triton X-100, supplemented with Complete antiprotease. Lysates were spun at $16,100 \times g$ for 15 min. The supernatant was recovered and diluted to 1 mg/ml. A small volume (50 µl of NeutrAvidin-coated agarose bead slurry (Thermo Scientific 29200) was prewashed two times in buffer A with 0.1% Triton X-100, and 500 µg of cell lysate was incubated with the beads for 2 hr with end-over-end rotation at 4°C. Beads were then washed five times in buffer A with 0.1% Triton X-100 for 5 min with end-over-end rotation at 4°C. Proteins were eluted from the beads by boiling in Laemmli sample buffer at 75°C for 5 min, and samples were analyzed by SDS–PAGE followed by immunoblot. qRT-PCR

Quantiative rt-PCR was performed as described in *Gokhale et al. (2015a)* and *Larimore et al. (2013)*). RNA from HEK293T cells was TRIzol-extracted (Invitrogen), and isolated RNA was reverse transcribed into cDNA using SuperScript III first strand synthesis (Invitrogen). PCR amplifications were performed on a LightCycler480 real time plate reader using LightCycler 480 SYBR Green reagents (Roche).

## Drosophila strains and procedures

The following strains were obtained from the Bloomington Drosophila Stock Center, Bloomington, Illinois: w[1118]. w[1118]; PBac{w[+mC]=RB}CG4848[e02840]/TM6B, Tb[1]; w[1118]; w[1118]; P{w[+mC]=Ddc-GAL4.L}Lmpt[4.36] and C155 GAL4 animals were from Bloomington (P{GawB}elav[C155], Fly Base ID FBti0002575). w[1118]; UAS-ATP7-wt was a gift of Richard Burke, Monash University, Australia.

Flies were reared on standard Molasses Food (Genesee Scientific) at 25C in 12 hr:12 hr light:dark cycle. Copper feeding toxicity experiments were performed as described using 5% glucose as control feed or 5% glucose supplemented with 1 mM $CuCl_2$ (*Gokhale et al., 2015a*).

Larval dissections, immunohistochemistry, and confocal microscopy were performed as described previously (*Gokhale et al., 2016*, *2015b*; *Mullin et al., 2015*). Wandering third-instar female larvae were dissected in normal HL3, fixed in 4% paraformaldehyde for 1 hr, and stained with HRP–FITC conjugated antibody for 2 hr at room temperature (1:500). An inverted 510 Zeiss LSM microscope was used for confocal imaging of synapses at muscle 6/7 on either the second or third segment.

Bouton counts and branch length were calculated using FIJI (*Schindelin et al., 2012*) with experimenters blind to the genotype of the animal.

## Statistical analysis
Experimental conditions were compared using Synergy Kaleida-Graph, version 4.1.3 (Reading, PA) or Aabel NG2 v5 $\times$ 64 by Gigawiz as specified in each figure.

## Acknowledgements

This work was supported by grants from the National Institutes of Health: NS088503 to VF, DK093386 to MP, GM083144 to VL, and the HERCULES grant (NIEHS: P30 ES019776). We are indebted to the Faundez lab members for their comments. Stocks obtained from the Bloomington *Drosophila* Stock Center (NIH P40OD018537) were used in this study. SR-R was supported by a fellowship from the Marion T. Clark Research Fund and William Joe Frierson Research Fund. We would like to thank Maria Olga Gonzalez Gonzalez for support during the manuscript figure making in Santiago, Chile and for providing mitochondria to VF.

## Additional information

### Funding

| Funder | Grant reference number | Author |
| --- | --- | --- |
| National Institute of Neurological Disorders and Stroke | NS088503 | Victor Faundez |
| National Institute of Diabetes and Digestive and Kidney Diseases | DK093386 | Michael Petris |
| National Institute of General Medical Sciences | GM083144 | Vladimir Lupashin |
| National Institute of Environmental Health Sciences | P30 ES019776 | Dana Boyd Barr |

The funders had no role in study design, data collection and interpretation, or the decision to submit the work for publication.

### Author contributions
HSC, Conceptualization, Data curation, Formal analysis, Supervision, Funding acquisition, Investigation, Visualization, Methodology, Writing—original draft, Project administration, Writing—review and editing; JM, SAZ, Conceptualization, Data curation, Formal analysis, Investigation, Methodology, Writing—original draft, Writing—review and editing; SR-R, Data curation, Formal analysis, Investigation, Methodology, Writing—review and editing; CH, AG, DBB, AV-M, Conceptualization, Resources, Data curation, Formal analysis, Investigation, Methodology, Writing—review and editing; JBB, EW, Conceptualization, Resources, Investigation, Methodology, Writing—review and editing; MP, Resources, Investigation, Writing—review and editing; PD'S, Resources, Formal analysis, Investigation, Writing—review and editing; PP, Conceptualization, Resources, Formal analysis, Investigation, Writing—review and editing; VL, Conceptualization, Resources, Data curation, Formal analysis, Supervision, Funding acquisition, Visualization, Methodology, Writing—original draft, Project administration, Writing—review and editing; VF, Conceptualization, Resources, Data curation, Formal analysis, Supervision, Funding acquisition, Investigation, Visualization, Methodology, Writing—original draft, Project administration, Writing—review and editing

### Author ORCIDs
Vladimir Lupashin, http://orcid.org/0000-0002-2350-1962
Victor Faundez, http://orcid.org/0000-0002-2114-5271

## Additional files

### Supplementary files

• Supplementary file 1. Reagents and proteomic findings from neuroblastoma cells. Tabs contain table of antibodies and primers used in this study. BCS and Cu tabs include all proteome data from crosslinked ATP7A complexes isolated from BCS treated and copper treated neuroblastoma cells as indicated in *Figure 1* and Material and methods. Cutoff selection criteria of hits are defined in Material and methods.

• Supplementary file 2. Proteomic findings from ATP7A null and rescue cells. Proteome data from crosslinked ATP7A complexes isolated from human ATP7A null fibroblasts and rescue cells (ATP7A$^{R/R}$) as indicated in *Figure 1* and Material and methods. Cutoff selection criteria of hits are defined in Material and methods. All hits below the selection cutoff were used to curate the BCS and copper treated proteomes in *Supplementary file 1* to give origin to the ATP7A interactome in the *Supplementary file 3* Tab (BCS+Cu Hits).

• Supplementary file 3. Curated proteins defining the ATP7A interactome and their analysis by bioinformatics. Selected hits from BCS treated cells and copper treated cell immunoisolated ATP7A complexes. Tab with the sum of these hits (BCS+Cu Hits) was used for bioinformatics (Tabs A-C). Crapome lists hits from one of the CRAPome datasets and the proteins shared by the ATP7A interactome and the CRAPome. Tabs (A), (B), and (C) contain DAVID, ENRICHR and GDA bioinformatic analyses, respectively, which are graphically depicted in *Figures 2* and *3*.

### Major datasets

The following datasets were generated:

| Author(s) | Year | Dataset title | Dataset URL | Database, license, and accessibility information |
|---|---|---|---|---|
| Victor Faundez | 2017 | ATP7A IP DSP treated cells after incubation with BCS or CuCl2 | http://www.peptideatlas.org/PASS/PASS01000 | Available at the PeptideAtlas database (dataset identifier PASS01000) |
| Victor Faundez | 2017 | ATP7A IP Menkes and Rescued Menkes Cells | http://www.peptideatlas.org/PASS/PASS01001 | Available at the PeptideAtlas database (dataset identifier PASS01001) |

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
