## [Decision Letter]

Thank you for submitting your article "The Interactome of the Copper Transporter ATP7A Belongs to a Network of Neurodevelopmental and Neurodegeneration Factors" for consideration by *eLife*. Your article has been reviewed by three peer reviewers and the evaluation has been overseen by Andrew West as the Reviewing Editor and Anna Akhmanova as the Senior Editor. The reviewers have opted to remain anonymous at this time.

The strengths of the study are the definition of an ATP7A interactome. This interactome is used to posit that the COG complex critically dictates copper uptake through ATP7A interaction. The interactome may provide a novel database for understanding metal transport in degenerative diseases. Further, the newly defined COG complex and ATP7A interaction may provide new targets for intervention in Menke's disease and other disease where metal transport is disrupted.

The reviewers have discussed the reviews with one another and the Reviewing Editor and identified two points that constitute essential revisions required for further consideration.

1) Transparency and rigor are key factors at *eLife*. As such, a compendium of easily accessible and annotated mass spectrometry data from all experiments used for analysis is required to be submitted as supplement. Further, source mass spectrometry (e.g., raw data files, xml, etc.) should be made available in the public domain (see a recent publication on the subject PMID 25721607). Reviewers would also like the 'interactome' filtered via known contaminants in IPs (see PMID 23921808). Finally, all raw data and bioinformatic analyses that compose the main figures need to be presented in full as supplemental material and in the statistical analysis section.

2) All three reviewers felt that the knockout experiments with COG and ATP7A mutation are not sufficient to demonstrate a role for the interaction of the two. In other words, only part of the proposed COG and ATP7A interaction has been studied and the "loop" needs to be closed with additional approaches. Reviewers suggested addressing COG toxicity by delivering copper to flies/cells (e.g., CuGTSM), or using other genetic and biochemical approaches to show the relationship between COG defects and ATP7A loss of function as a primary driver of toxicity. Related to this, the specificity of the COG complex activity toward copper transporters over other transporters like zinc metal could be strengthened using the tools already developed.

At the very least, a rescue of the COG-deficient flies with restoration of copper levels would represent a compelling addition to this story. This would strengthen the conclusion of a direct connection between copper and COG complex, although, there might be technical problems with this experiment that I am unaware of. This may be enough to show the transporter is a direct cargo of COG, however one of the main conclusions would have to drop out, namely that copper deficiency is a driving feature of COG complex defect-induced neurotoxicity.

---

## [Author Response]

*1) Transparency and rigor are key factors at eLife. As such, a compendium of easily accessible and annotated mass spectrometry data from all experiments used for analysis is required to be submitted as supplement. Further, source mass spectrometry (e.g., raw data files, xml, etc.) should be made available in the public domain (see a recent publication on the subject PMID 25721607).*

We have now included [Supplementary-material SD1-data]–[Supplementary-material SD3-data] that contain all mass spectrometry data from three independent experiments. Concerning source MS files, we will make them available through http://www.peptideatlas.org/, which is suggested in PMID 25721607. File submission asks for publication PMID or title-journal information. Currently, we are contacting service providers to gain access to these files.

*Reviewers would also like the 'interactome' filtered via known contaminants in IPs (see PMID 23921808).*

This is a great suggestion that further emphasizes the confidence in our interactome. We tested the ATP7A interactome with a CRAPome dataset that resembles our experimental conditions. The CRAPome dataset is similar to either the BCS or the copper ATP7A interactomes in its number of hits (see revised Figure 2 for Venn diagrams). Importantly, the overlap of our experimentally curated ATP7A interactome with the CRAPome is just 31 proteins, or a 5.7% of the totality of the curated ATP7A interactome. This indicates that our experimental approach to define a curated ATP7A interactome carefully eliminated proteins that spuriously bind to beads-antibody matrices. We attribute this success to the curation of the ATP7A interactome with our dataset generated with ATP7A null cells (revised Figure 1 and [Supplementary-material SD2-data]–[Supplementary-material SD3-data]), which in essence is our own ‘CRAPome’.

The 31 proteins overlapping between the CRAPome and the ATP7A interactome may be real interactors. Thus, we decided to leave them in the final ATP7A interactome dataset. Importantly, eliminating these 31 proteins neither affect our bioinformatic analyses nor they compromise our experimental choices for study. We now provide the list of these 33 proteins in [Supplementary-material SD3-data] for readers to judge. We hope reviewers will agree with our rationale and decision to keep these 31 proteins as part of the ATP7A interactome.

*Finally, all raw data and bioinformatic analyses that compose the main figures need to be presented in full as supplemental material and in the statistical analysis section.*

We included these data and now they are part of [Supplementary-material SD1-data]–[Supplementary-material SD3-data].

*2) All three reviewers felt that the knockout experiments with COG and ATP7A mutation are not sufficient to demonstrate a role for the interaction of the two. In other words, only part of the proposed COG and ATP7A interaction has been studied and the "loop" needs to be closed with additional approaches. Reviewers suggested addressing COG toxicity by delivering copper to flies/cells (e.g., CuGTSM), or using other genetic and biochemical approaches to show the relationship between COG defects and ATP7A loss of function as a primary driver of toxicity.*

This is a very important suggestion that we addressed in revised Figure 9 and 10. These data provide genetic evidence of interactions between ATP7A and COG complex subunits in the developing synapse and adult dopaminergic neurons in *Drosophila*. In particular, revised Figure 10 demonstrates that a genetic interaction between ATP7A and COG is copper-dependent.

We previously demonstrated that feeding copper in the diet of adult flies is lethal in wild type animals of both sexes (PMID: 26199316). We reasoned that ATP7A overexpression in *Drosophila* dopaminergic neurons should be protective of copper feeding induce lethality because ATP7A overexpression cell-autonomously decreases cellular levels of copper. This effect is due to ATP7A mistargeting to the cell surface as reported by others (see references Lye et al., 2011 and Lin and Goodman, 1994 in revised manuscript). We selected dopaminergic neurons because of their relevance to Parkinson disease, a disease associated by bioinformatics to the ATP7A interactome (revised Figure 3 and [Supplementary-material SD3-data]). We hypothesized that protective phenotypes induced by genetically increasing ATP7A expression in neurons should be modulated by loss-of-function mutations in the COG complex, which decreases ATP7A expression in human cells (revised Figure 6). This model was thoroughly tested and its predictions fulfilled, thus offering a powerful genetic argument for ATP7A and COG participating on the same copper homeostasis pathway.

*Related to this, the specificity of the COG complex activity toward copper transporters over other transporters like zinc metal could be strengthened using the tools already developed.*

This is an interesting suggestion. The human genome encodes 10 SLC30 type and 14 SLC39 type zinc transporters that have diverse expression patterns and topologies of zinc transport towards or away from the cytoplasm PMID: 24745988. *Drosophila* has 2 SLC30 and 2 SLC39 zinc transporters PMID: 24063361. Because of this diversity of transporters and zinc transport topologies, it is hard to assess the consequences of COG deficiency on zinc transporters since the selection of candidate(s) for study is so big. Currently, we do not have mammalian or *Drosophila* tools to assess these transporters. However, our data show that COG genetic defects negligibly affect zinc cellular content (see revised Figure 7). We would like ask for a waiver for experiments addressing zinc transporters for the reasons above indicated.